

# Laboratory Study of the Properties of Frazil Ice Particles and Flocs in Water of Different Salinities

Christopher C. Schneck[1], Tadros R. Ghobrial[1], Mark R Loewen[1]

[1]Department of Civil and Environmental Engineering, University of Alberta, Edmonton, T6G 1H9, Canada

*Correspondence to*: Tadros R. Ghobrial (triad@ualberta.ca)

**Abstract.** Measurements of the size and shape of frazil ice particles and flocs in saline water and of frazil ice flocs in freshwater are limited. This study consisted of a series of laboratory experiments producing frazil ice at salinities of 0 ‰, 15 ‰, 25 ‰, and 35 ‰ to address this lack of data. The experiments were conducted in a large tank in a cold room with bottom mounted propellers to create turbulence. A high-resolution camera system was used to capture images of frazil ice particles

and flocs passing through cross polarizing lenses. The high-resolution images of the frazil ice were processed using a computer algorithm to differentiate particles from flocs and determine key properties including size, shape, and volume. The size and volume distributions of particles and flocs at all four salinities were found to fit lognormal distributions closely. The concentration, mean size, and standard deviation of flocs and particles were assessed at different times during the supercooling process to determine how these properties evolve with time. Comparisons were made to determine the effect of

salinity on the properties of frazil ice particles and flocs. The overall mean size of frazil ice particles in saline water and freshwater was found to range between 0.52 and 0.45 mm with particles sizes in freshwater ~ 13% larger than in saline water. However qualitative observations showed that frazil ice particles in saline water tend to be more irregularly shaped. The overall mean size of frazil ice flocs in freshwater was 2.57 mm compared to a mean size of 1.47 mm for flocs in saline water. Estimates for the porosity of frazil ice flocs were made by equating the estimated volume of ice produced based on

thermodynamic conditions to the estimated volume of ice determined from the digital images. The estimated porosities of frazil ice flocs were determined to be 0.86, 0.82, 0.80, and 0.75 for 0 ‰, 15 ‰, 25 ‰, 35 ‰ saline water, respectively.

## 1 Introduction

Frazil ice particles are small crystals that form when water is supercooled (i.e. cooled below 0°C) and turbulent. Given these conditions, frazil ice production and growth are naturally occurring processes that may be observed in rivers and oceans

(Martin, 1981). In northern rivers (i.e. freshwater), the individual frazil ice particles are transported by the turbulent flow and begin to collide with one another. These collisions and the adhesive properties of the ice cause them to sinter together into groups of particles known as frazil flocs in a process known as flocculation. Sintering is the process by which particle and flocs bind together and adjust their shape in order to minimize free surface energy (Hobbs, 1970). Once these frazil flocs reach a mass large enough for their buoyancy to overcome the entraining turbulence of the flow, the flocs will rise to the



surface of the river. At the surface, the flocs continue to combine and form slush that then freezes together due to the cold air to form frazil pans or pancake ice (Hicks, 2009). The pans will move with the flow of the river and continue to collide and combine with other pans to form larger ice floes known as rafts. Once a high concentration of the rafts and pans has been reached on the river, a congestion or arrest of incoming ice rafts and pans will occur in a process known as bridging (Hicks,

2009). Bridging often occurs where there is a flow constriction (i.e. at a bridge or island) or at a bend in the river. Incoming ice from upstream will continue to be arrested at the bridging location as the solid ice cover propagates upstream (Beltaos, 2013). Once the solid ice cover has been formed, the layer of ice provides insulation from the cold air to the water flowing beneath, thereby preventing supercooling of the water column and preventing frazil ice production for the remainder of the winter (Beltaos, 2013).

In oceans (i.e. saline water), similar conditions of supercooling and turbulence are necessary for frazil ice production to begin. In polar regions, there are two common sets of conditions that produce different types of ice cover in the ocean (Weeks, 2010). The first is calm winds, cold air temperatures, and little swell, and in this case, frazil ice starts to form and freeze together into a continuous skim ice on the sea surface. The second set of conditions that produce frazil ice are cold air

temperatures with appreciable swell, waves, wind, and blowing snow (Weeks, 2010). As ice production begins, frazil ice forms a slurry layer on the surface and as more ice is added to the layer, the viscosity of the layer continually increases. This layer of soupy, viscous ice that forms is called grease ice. Grease ice is very flexible and weak and would not hold its shape if removed from the water. Under conditions of heavy snow, a snow water mixture can form on the surface of the ocean, which is indistinguishable from grease ice. Pancake ice, similar to rivers, has also been observed in the ocean (Weeks, 2010).

Frazil ice particles have been observed to deposit under sea ice, grow in situ and contribute to its thickness (Langhorne, et al., 2015). These crystals known as platelet ice are characterized as fragile and dendritic in shape (Weeks, 2010). Frazil ice was also observed to form in gaps in the ice pack (called leads) and in extensive areas of open water (called polynyas), as observed by (Skogseth et al., 2009). At the mouths of rivers, the salinity difference between the ocean and the river can lead to frazil ice production (Martin, 1981). The temperature difference between the saline water and freshwater can lead to

supercooling of the freshwater resulting in the production of frazil ice. Polynyas play an important role in ice production and brine production in the ocean (Weeks, 2010). As frazil ice forms in a polynya, the salt is rejected by the ice resulting in higher salinity which depresses the freezing point even further and results in colder water in the vicinity of the polynya. Cavalieri & Martin (1994) found that the rejection of salt during frazil ice production in polynyas in the Canadian Arctic contribute to the cold salty water in the Arctic halocline. In the polynya located in the Chukchi Sea off Cape Lisburne and

Point Hope, Alaska, rapid ice growth occurs and is accompanied by an ocean salt flux that produce a dense outflow on the ocean floor (Martin, 1981). Frazil ice may also contribute to the transport of nutrients and metals which may alter the water quality (De Santi and Olla, 2017).



Several laboratory and field studies have reported measurements on the size and shape of individual frazil ice particles in freshwater. The reported frazil particles were dominantly disc-shaped with a range of diameters between 23 μm and 5 mm, and particle sizes fitting a lognormal particle size distribution in laboratory (Gosink and Osterkamp, 1983; Daly and Colbeck, 1986; Ye et al., 2004; Clark and Doering, 2006; Ghobrial et al., 2012; McFarlane et al., 2015) and field studies
(Gosink and Osterkamp, 1983; Richard et al., 2011; Ghobrial et al., 2013; Marko et al., 2015; McFarlane et al., 2017). A limited number of measurements of the shape and size of individual frazil ice particles in saline water have been performed, and to the best of the authors' knowledge, no studies have reported particle size distributions. Laboratory experiments performed by Kempema et al. (1993) reported disc diameters of 1 to 5 mm in fresh water and 1 to 3 mm in water at salinities of 29.14 ‰ and 32.00 ‰. In lab experiments with salinities between 36 ‰ and 38 ‰, Smedsrud (2001) reported an average
diameter of irregular shaped frazil ice particles of 2 mm. Disc shaped particles measuring approximately 1 mm in diameter were reported in laboratory experiments at a salinity of 35.5 ‰ by Martin and Kauffman (1981). Ushio and Wakatsuchi (1993) performed laboratory experiments investigating the effect of wind conditions and salinity on the properties of frazil ice particles and found that under windy, high salinity conditions dendritic crystals having diameters of 2 to 3 mm were produced, and under calm air, lower salinity conditions disc shaped crystals with diameters of approximately 5 mm were
produced. In summary, in saline water, the individual frazil ice particle sizes reported are comparable to freshwater observations, and there is a consensus that individual frazil ice particles are more irregularly shaped in saline water.

A small number of studies have been performed to investigate the process of flocculation and the properties of frazil ice flocs. Floc properties such as porosity and size are needed to estimate the mass and the volume of frazil deposit (Park and
Gerard, 1984). Experiments suggest that frazil ice in saline water is less adhesive than freshwater frazil ice, resulting in less tendency to flocculate (Hanley and Tsang, 1984; Kempema et al., 1993). Hanley and Tsang (1984) performed laboratory experiments in a tank with turbulence created by a propeller with 44 ‰ water. They observed that saline water flocs would disperse and break apart when passing near the propeller and that at the surface the frazil had no detectable resistance to force applied by a finger. This was attributed to the salt rejection that occurs when the frazil ice is formed in saline water. As
the salt is rejected by the frazil ice, it forms a thin layer of slightly higher salinity water around the frazil ice particle (Hanley and Tsang, 1984). The freezing point of the saline water around the particle will then be slightly reduced and this inhibits the ability of the ice to grow and adhere to neighbouring particles and flocs. Kempema et al. (1993) created flocs in a racetrack flume powered by propellers in a walk-in freezer. The study described the flocs as a group of particles aligned with their flat surfaces in contact. The flocs in saline water were also observed to be smaller on average and more dendritic in appearance
when compared to freshwater flocs (Kempema et al., 1993). Saline water flocs were observed to stay suspended in the flow and did not rise to the surface until the experiment was stopped and the turbulence subsided. Park and Gerard (1984) conducted laboratory experiments on artificial frazil flocs fabricated from plastic discs (diameter of ~10 mm and thickness between 0.16 and 0.41 mm) to assess the hydraulic characteristics of frazil flocs. They fabricated artificial flocs ranging in size between 10 and 100 mm, and in porosity between 0.01 and 0.89 and estimated drag coefficients from fall velocities



measured in a tank. It was found that the drag coefficient of flocs of disk-like particles was significantly higher than that for a sphere of the same bulk density. In Clark and Doering's (2009) freshwater experiments in a counter rotating flume, measurements of the size of frazil ice flocs were made. In the study, a floc was defined as a group of particles that have sintered together, however, only objects with major axis larger than 17 mm were considered flocs. They found that higher

levels of turbulence intensity tended to constrain the formation of large frazil flocs.

In summary, in freshwater there have been a number of laboratory studies investigating frazil ice particles but very few focused on frazil flocs. However, in saline water, there have been very few studies of frazil ice particles or frazil flocs. In this study laboratory measurements of frazil ice particles and flocs in water at salinities 0 ‰, 15 ‰, 25 ‰, and 35 ‰ were

conducted. Measurements of the size and shape of individual frazil ice particles and frazil ice flocs in saline and freshwater can be applied to improve river ice models (e.g. Shen, 2010) and sea ice formation models (e.g. Rees Jones and Wells, 2018; De Santi and Olla, 2017).

## 2 Experimental Set-up and Methods

Experiments were performed in the frazil ice production tank in the University of Alberta's Cold Room Facility as described

in Ghobrial et al. (2012). An image of the tank and the experimental set-up is presented in Fig. 1. The tank has a base dimension of 0.8 m by 1.2 m and was filled to a depth of 1.2 m. The four bottom-mounted propellers used to generate turbulence in the tank are driven by four NEMA 34 DC variable speed electric motors (278 W, 1.514 N-m of torque, max speed of 1750 rpm). The turbulence intensity was held constant by keeping the propeller speed constant at 325 rpm for all experiments. The speed of each motor was verified using a laser tachometer. In a similar series of experiments in the same

tank, McFarlane et al. (2015) found that the tank-averaged turbulent kinetic energy dissipation rate was 336 $cm^2/s^3$ at a propeller speed of 325 rpm, and this fell within the range of dissipation rates estimated for rivers in Alberta.

Two types of light-emitting diode (LED) lights were used in the experiments to illuminate the frazil ice particles: a Genaray SpectroLED Essential 360 Daylight LED Light (3,200 lux at 1.0 m, 360 LED bulbs, 29.8 cm by 29.8 cm) and an Andoer

FalconEyes RX-18TD 504 pcs LED Light (3660 lux at 1 m, 504 LED bulbs, 70.0 cm by 46.0 cm). The light source was placed on the far side of the tank (Fig. 1). On the opposite side of the tank, two Cavision glass polarizers were mounted in the tank flush to the front glass wall. Two different polarizer sizes were used in the experiments, a 10 by 10 cm and a 16 by 16 cm polarizers. In both cases, the polarizers were installed at 90° to one another in order to cross polarize the light. This setup produced a black background in the captured images where only the ice particles / flocs passing between the two

polarizers, that had refracted the incident light, were visible. The polarizers were mounted as close as possible to the front glass sidewall to prevent any distortion of the images caused by suspended frazil ice getting between the sidewall and the polarizers. Digital images of frazil ice and flocs were captured using a Nikon D800 camera (36-megapixel resolution)



equipped with an AF Micro-Nikkor 60 mm f/2.8D lens. The camera was positioned outside of the tank and focused on the polarizers (see Fig. 1). A space heater was used to blow hot air against the outside of the glass sidewall to prevent frost formation.

Based on some preliminary experiments, three different camera and polarizer set-ups that provided the best quality images, with regards to brightness and clarity were determined and these are summarized in Table 1. For Set-up 1, it was determined that in freshwater the 2.2 cm spacing between the polarizers was appropriate to capture individual frazil particles but prevented many flocs from advecting between the polarizers and also the flocs were sometimes too large for the 47.5 by 31.7 mm field of view. For Set-up 2, the flocs in saline water were observed to be small enough that a 2.2 cm spacing between the
polarizers did not restrict their movement and the flocs were small enough to fit in the slightly larger field of view of 61.3 by 40.9 mm. In the case of Set-up 3, a 3.5 cm spacing between the polarizers and a larger field of view of 162.9 by 141.3 mm were needed to accommodate the larger freshwater flocs. The three different set-ups were used during the five series of experiments listed in Table 2.

The temperature of the water in the tank during the experiments was recorded at a rate of 0.62 Hz using a Sea-Bird SBE 39 temperature recorder (accuracy of ± 0.002 °C). The temperature sensor was mounted in the tank and placed at the approximate centre of the tank. Temporal variations of air temperature in the cold room were measured using RBR Solo Temperature Loggers (accuracy of ± 0.002 °C) at a frequency of 1 Hz. A series of experiments were performed in freshwater to determine if the water in the tank was well mixed and if the water temperature was uniform in the tank at a propeller speed
of 325 rpm. In addition to the Sea-Bird temperature sensor placed approximately in the middle of the tank, six RBR Solo temperature loggers were placed at different locations throughout the tank. During each experiment, the water in the tank was cooled until frazil ice particles were generated and then the experiment was stopped. The temperature difference between each RBR Solo and the Seabird was computed at each time step throughout the experiment and these differences were then averaged over the entire event duration. The mean differences ranged from - 0.00337 °C to 0.00482 °C indicating
that the temperature was approximately uniform. It is noteworthy that the average temperature difference between the Sea-Bird at the centre of the tank and the RBR Solo at the location of the polarizers is 0.00067 °C, which is less than the accuracy of the RBR Solos and Sea-bird (± 0.002 °C). This indicated that the temperature measurements taken at the centre of the tank using the Sea-bird are reflective of the conditions at the location where the frazil ice images were taken. It is also important to note that the observed temperature differences are approximately an order of magnitude smaller than the
maximum supercooling temperatures observed in this study.

The tank was filled with fresh, filtered tap water to a depth of 1.2 m for all freshwater and saline water experiments. Saline water experiments were performed at salinities of 15 ‰, 25 ‰ and 35 ‰. A salinity of 35 ‰ was chosen because it is near the upper limit of salinities found in the ocean. Intermediate salinities of 15 ‰ and 25 ‰ were chosen as they could be





representative of salinities at salt-freshwater interfaces such as estuaries. Furthermore, by choosing intermediate salinities, the change in frazil ice properties as a function of salinity could be assessed. The required mass of salt to attain each salinity concentration was calculated and measured using a digital scale with accuracy of 0.2 g. Sifto Hy-Grade Food Grade Salt was used and is specified to be predominately sodium chloride (99.77 % to 99.91 % NaCl). Due to evaporation filtered tap water

was periodically added to the tank to maintain the 1.2 m depth.

At the start of an experiment the propellers were turned on, the Sea-Bird temperature logger was started and the polarizers were placed in the tank at the specified spacing. Images of a ruler positioned at the front, back, and midpoint between the polarizers were captured prior to each experiment to measure the field of view. Ten background images of the water in the

tank prior to any ice formation were also captured for each experiment. Next, the cold room was programmed to reduce the air temperature from approximately +2 °C to -8.02 °C ± 0.20. The camera was then programed to capture images at a rate of 1 Hz starting at approximately the time when the water had cooled to the freezing point. Images were capture for a total duration of 999 s and 1998 s in freshwater and saline water, respectively.

## 3 Freezing Point Experiments

Supercooling curves are defined as time series plots of temperature during the time period when the temperature is below the freezing point. An idealized supercooling curve for freshwater assuming a constant heat loss is presented in the upper plot of Fig. 2. Initially the temperature decreases linearly with time due to the constant rate of heat loss but as time increases the slope of the curve (i.e. the rate of temperature decrease) decreases due to the heat released when frazil ice begins to form. The latent heat of fusion and the mass of ice created per unit time determine how rapidly the slope of the curve decreases,

and at some point, the temperature reaches its minimum value (i.e. zero slope) defined as the maximum supercooling temperature. After this point in time, the temperature continues to increase as more frazil ice is formed, and the curve eventually reaches a constant residual temperature which is typically slightly below the freezing temperature if frazil ice production continues to occur (Hanley and Tsang, 1984). The principal supercooling period is defined as the time from when the water is cooled below the freezing temperature until it reaches the residual temperature. In saline water, the supercooling

curves follows a slightly different pattern when compared to freshwater, as illustrated in the lower plot in Fig. 2. The shape of the curve and the physics of the process are the same as in freshwater up until the time the residual temperature is reached. In saline water during the residual phase of the supercooling curve, ice continues to form causing the salinity of the water to continuously increase slowly. As a result, the residual temperature does not remain constant but decreases slowly (Brescia et al., 1975).

It was essential to determine the freezing point of water at the different salinities tested in these experiments. For this purpose, experiments were conducted to measure the freezing point at the three salinities and it was assumed to be 0 °C in





freshwater. The experiment consisted of sampling approximately 1000 mL of water from the tank during each experiment at a given salinity. This water sample was then placed in the cold room during the experimental run and mixed using a magnetic mixer. The water temperature in the stirred beaker was recorded using an RBR Solo temperature sensor. Supercooling curves for each of the freezing point depression experiments were plotted and the linearly sloping portion of

the residual was extrapolated back to where it intersects the curve as illustrated in Fig. 2. Mair et al. (1941) developed this method and though it does not provide the exact freezing point, it has been shown to produce a value accurate to within a few percent (She et al., 2016). The freezing point measurements made during each repeated experiment at a given salinity were averaged and the mean freezing point (± standard deviation) for salinities of 15 ‰, 25 ‰, and 35 ‰ was -0.89 ± 0.020 °C, -1.48 ± 0.019 °C and - 2.09 ± 0.023 °C, respectively.

**4 Repeatability of Experiments**

Repeated runs of experiments were conducted during each of the five series of experiments in order to reduce the uncertainty in the results by ensemble averaging. In total, 62 experiments were conducted. Anomalous (i.e. outlier) runs (typically due to inconsistent air temperatures in the cold room, images out of focus, or frost build-up on the tank glass) were discarded from the analysis. This process eliminated 16 experiments and there were 9 to 10 repeated experiments that could be ensemble

averaged for each series of experiments. In Fig. 3 temperature time series from the 9 repeated runs conducted at a salinity of 15 ‰ are plotted. The supercooling curves aligned quite well, indicating that the experimental conditions in the cold room were adequately controlled and produced repeatable experiments. Note that a negative slope was not observed during the residual of the freezing point depression experiments in saline water as was illustrated in Fig. 2. This was because the quantity of ice produced in the tank relative to the total volume of water in the tank was not large enough to reduce the

freezing point significantly. A summary of the statistics of the temperature time series during the five series of experiments is presented in Table 2. The maximum supercooling, defined as the minimum temperature minus the freezing point, and the cooling rates ranged between -0.0752 °C and -0.0924 °C, and 0.0092 and 0.0123 °C/min, respectively. The COV of the maximum supercooling and the cooling rates ranged between 2.06 and 4.57%, and 2.87 and 7.11%, respectively confirming the experiments were repeatable within acceptable limits.

**5 Data Processing**

**5.1 Raw images**

The raw images of the frazil ice particles and flocs from each experiment were visually examined to assess the qualitative characteristics of the freshwater and saline water frazil ice particles. This visual analysis helped guide the development of the image processing algorithm used to compute the properties of frazil ice particles and flocs. In Fig. 4, a pair of representative

raw images a salinity of 35 ‰ and in freshwater are shown to illustrate the qualitative differences between frazil ice particles



and flocs in saline and freshwater. An image at salinity of 35 ‰ is presented but the qualitative characteristics of the particles and flocs at this salinity are similar to those observed at salinities of 15 ‰ and 25 ‰. Figure 5 presents a zoomed view of the different particles' shapes observed in the experiments. In freshwater experiments, the typical shape of the particles was very consistent with previously reported studies in that the individual frazil ice particles were predominantly

disc-shaped. In saline water experiments, the shapes of individual frazil ice particles were a combination of disc, dendritic, hexagonal and other irregular shapes as shown in Fig. 5.

The average size of frazil particles appeared to be larger in freshwater than in saline water experiments. Also, in freshwater frazil ice flocs were remarkably bigger than those in saline water and tended to rise to the surface even when the turbulence

created from the propellers was still present, while in saline water, flocs remained in suspension until the propellers were turned off. In saline water, there are many smaller flocs comprised largely of irregularly shaped particles. In stark contrast, in the freshwater image (Fig. 4b), there are fewer smaller flocs present and one very large floc that dominated the image. The very large freshwater floc is an order of magnitude larger than the largest floc visible in saline water image, and it is evident that freshwater flocs are comprised largely of disc shaped particles. There were no significant differences observed when

comparing the particles and flocs sizes at the different salinities.

### 5.2 Image processing

Images from each experiment were analysed to compute the properties of individual frazil ice particles and frazil ice flocs using a modified image processing algorithm originally developed by McFarlane et al. (2015). Modifications to the algorithm were made to more accurately determine particle properties for the experimental conditions in this study and to

distinguish between individual frazil ice particles and frazil ice flocs. First the average of ten background images (before frazil ice production starts) were subtracted from each image in the series. The raw images from the series were then converted to grayscale by eliminating the hue and saturation information while retaining the luminance. The grayscale images were then converted to a binary image using a simple threshold. The threshold value is specified as a scalar luminance value between 0 and 1. As such any pixel with a scalar luminance value less than the specified threshold would be

assigned a value of 0 (black) and any pixel with a scalar luminance value greater than the specified threshold would be assigned a value of 1 (white). At this point in the algorithm, the white objects correspond to frazil ice particles and/or flocs, and the black pixels correspond to the dark ice-free background. Any objects that were touching the border of the image were eliminated from the binary image to prevent properties of portions of individual frazil ice particles and flocs from being computed. Finally, the white objects in the binary images were then dilated and eroded by five pixels to ensure any possible

holes in the white objects had been filled in. Next, the objects that represent individual frazil ice particles or flocs were analysed for key properties such as the area, perimeter, and the centroid. In addition, the major and minor axis lengths, and eccentricity of a fitted ellipse that has the same second moments as the detected object were also computed. The eccentricity $e$, of an ellipse is calculated as follows:



$$e = \sqrt{1 - \frac{b^2}{a^2}} , \tag{1}$$

where $a$ and $b$ are semi-major and semi-minor axis length of the fitted ellipse of a given particle or floc, respectively. Figure 6 presents a binary image with superimposed fitted ellipses over detected frazil particles or flocs. It can be seen in Fig. 6 that the major axes of the ellipses do provide an accurate estimate of ice particle and floc sizes. Therefore, the major axis length

of the fitted ellipse is reported as the size of the frazil ice particles and flocs throughout the analysis following Clark and Doering (2009).

One of the objectives of the algorithm was to determine whether an object was an individual particle, or a floc. A frazil ice particle (both disc-shaped and irregular shaped particles) is more elliptical in shape than a frazil ice floc. Based on this

hypothesis, the algorithm compares the perimeter and area of a detected object to the perimeter and area of the corresponding fitted ellipse. If the area of the object was greater than 90 % of the corresponding fitted ellipse area, or the difference between the object perimeter and the fitted ellipse perimeter was less than 15% the object was identified as an individual frazil particle. The thresholds for perimeter and area were optimized by varying them to determine the values that correctly identified the highest percentage of particles and flocs. By manually checking approximately 500 individual frazil ice

particles and frazil ice flocs, it was found that the area and perimeter thresholds of 90 % and 15 % correctly identify objects 90 % and 94 % of the time in images taken in saline and freshwater, respectively. However, when this algorithm was applied to images of frazil ice captured using the Set-up 3 (to capture freshwater flocs using larger field of view) it was found to have an accuracy of only 86 %. This is likely because the pixel resolution was only 28.8 μm resulting in small particles looking more irregular in shape (i.e. pixelated). As a result, an additional criterion was developed to improve the accuracy of

the algorithm. This criterion utilizes the ratio of the area-over-perimeter of the objects (flocs or particles) to the area-over-perimeter of the corresponding fitted ellipses. If the object is a perfect ellipse, then this ratio equals one. As the shape of an object diverges from its fitted ellipse (i.e. more irregular shaped) this ratio becomes larger than one. A threshold for this ratio was used to distinguish between particles and flocs. Raw images of 500 objects were visually identified as particles or flocs and then compared to the algorithm predictions at various thresholds. It was found that a threshold ratio of 1.1 correctly

categorized the objects 95 % of the time. That is, if the ratio of an object's area-over-perimeter to its fitted ellipse's area-over-perimeter is greater than 1.1, then the object was identified as a frazil ice floc. Another benefit of using the properties of a fitted ellipse is that it provides information about the volume of ice particles or flocs from the volume of a fitted ellipsoid $V$ calculated as follows,

$$V = \frac{4}{3}\pi abc , \tag{2}$$

where $c$ is the dimension of a floc perpendicular to the plane of the image which is unknown. It was assumed that c was equal to the average of $a$ and $b$ but not greater than the distance between the polarizers when using Eq. (1) to estimate floc volume. When estimating the volume of an individual particle, it was assumed that the particles are approximately disc shaped and that the diameter to thickness aspect ratio of the individual particles was equal to 37, which was the mean aspect





ratio obtained by McFarlane et al. (2014). In all experiments it was found that the total volume of frazil ice particles was insignificant compared to the total volume of frazil ice flocs. Therefore, the volume of frazil ice particles was neglected when computing ice volumes.

## 6 Data Analysis

Results from the image processing algorithm were used to assess how the properties of the flocs and particles changed throughout a supercooling event. The particle/floc concentration as well as the mean particle/floc size and standard deviation were computed for each image throughout an experiment. The noise in the resulting time series was reduced by smoothing using a 35 s moving average. These smoothed times series of particle and floc properties (concentration, mean size, and standard deviation) were plotted for each repeated experiment conducted at the four salinities. The time series were

synchronized by aligning all the time series from each set of experiments at the time when the minimum temperature occurred. Next, the ensemble average time series of the particle or floc concentration, mean size, and standard deviation were calculated. In order to compare the ensemble average time series at the different salinities they were plotted as a function of a dimensionless time. The dimensionless time is defined as the time divided by $t_c$ defined as the time interval between when the freezing point and maximum supercooling occurred, as illustrated in Fig. 7. The time of freezing $t_f$ is taken

as the start of the experiment (i.e. $t/t_c = 0$), and $t_m$ corresponds to a dimensionless time $t/t_c = 1.0$. In all cases, by the time when $t/t_c = 2.0$, the supercooling curve had reached the residual temperature, and therefore the experiment was considered finished at this point in time.

Three time-phases were defined to assist in comparing results at different times in the supercooling process. The three phases

are illustrated in Fig. 8 and they apply to the particles and flocs experiments. Phase 1 is defined as the time from when the water reached the freezing temperature ($t/t_c = 0$) to the time when the water reached the maximum supercooling temperature ($t/t_c = 1.0$). The other two phases were defined relative to when the peak number of particles or flocs were observed in the ensemble-averaged time series following Clark and Doering (2006). Phase 2 is defined as the time from $t/t_c = 1.0$ to the time when 90 % of the peak number of particles or flocs is reached ($t_{90a}$). Phase 3 is defined as the time from $t_{90a}$ to the time when

90% of the peak number of particles or flocs is reached on the other side of the peak ($t_{90b}$). These three phases were well defined for individual particles experiments at all four salinities, and for the freshwater flocs experiments. For the saline water flocs experiments, the number of flocs reached a peak and then remained at a value above 90% of the peak for the remainder of the time interval and therefore $t_{90b}$ does not exist in this case. To consistently compare the results from the freshwater and saline flocs experiment, the Phase 3 dimensionless time interval from the freshwater flocs experiments was

used to estimate the results for Phase 3 for the saline water flocs experiments.



The volume of flocs estimated from the image processing algorithm was used to estimate the volume concentration of ice produced at the end of the principal supercooling. The volume of the fitted ellipsoid was multiplied by a factor that accounts for the porosity of the ice. Ghobrial (2012) conducted slush layer experiments in the same tank and estimated the slush porosity to be between 0.80 and 0.85. Also, Beltaos (2013) suggested a porosity of 0.80 may be a reasonable estimate.

Therefore, the volume of ice contained in a floc was estimated by multiplying the volume of its fitted ellipsoid by 0.20. A set of 25 images starting at the end of the principal supercooling were analyzed to get an average ice volume at this point in time for each experiment. The volume of the field of view was then calculated based on the determined pixel sizes and space between the polarizers for each set-up (see Table 1). The frazil ice volume concentration was estimated by computing the ratio of the volume of ice to the volume of the field of view. Finally, the shape of a floc (i.e. more elongated or more

circular) was quantified by computing the eccentricity of its fitted ellipse. Ellipses have eccentricity between zero and one, where an eccentricity of zero corresponds to a circle and an eccentricity of one corresponds to a straight line.

## 7 Results

### 7.1 Time series

Figures 9 and 10 present an example of the synchronized time series generated for the frazil particles and flocs, respectively,

for the 15 ‰ set of experiments. In these figures, the particles and flocs concentrations are calculated in terms of the number (N) of particles/flocs per unit volume ($cm^3$). Although only time series of the 15 ‰ experiments are shown, the trends in all experiments was similar in that the time series from all experiments line up together very well confirming the repeatability of the experiments. The time series from the different experiments at each salinity were then ensemble averaged for the particles and flocs as shown in Fig. 11 and 12.

In Fig. 11 the ensemble averaged time series of frazil ice particle concentration, mean particle size, and standard deviation at all salinities are plotted versus non-dimensional time ($t/t_c$). Figure 11a shows that at all four salinities, the particle concentration initially increased, reached a peak, decreased, and then asymptotically approached a constant value. In freshwater the peak particle concentration was 2.1 $cm^{-3}$ at $t/t_c = 1.20$, and in saline water the peaks were 2.2, 3.2, and 3.6 $cm^{-3}$

at $t/t_c = 1.28$, 1.24 and 1.23 at 15 ‰, 25 ‰ and 35 ‰, respectively. The 35 ‰ curve had the highest particle concentration throughout the entire time interval, and the freshwater curves had the lowest particle concentration throughout nearly the entire time interval, except near its peak where the freshwater concentration was briefly greater than the 15 ‰ concentration. In Fig. 11b the mean individual frazil ice particle size initially increased, reached a peak, decreased, and then reached an asymptotic value of approximately 0.5 mm in all four cases. The average growth rate of frazil particle was estimated for the

initial growth period and was found to be 0.174, 0.070, 0.033, and 0.024 mm/min for the 0 ‰, 15 ‰, 25 ‰, and 35 ‰, respectively. The maximum mean particle size for the three higher salinities occurred prior to when the maximum size was reached in freshwater. The maximum mean particle size occurred at $t/t_c = 0.96$, 0.81, 0.79, and 0.88 for 0 ‰, 15 ‰, 25 ‰,



and 35 ‰, respectively. The maximum mean size in freshwater was the largest at 0.68 mm, followed by 0.63, 0.57, and 0.53 mm for 15 ‰, 25 ‰ and 35 ‰, respectively. Figure 11c shows that the time series of the standard deviation are very similar in saline water where it increased from approximately 0.1 mm to 0.2 mm and reached a constant value of approximately 0.04 mm at approximately $t/t_c = 1$ until the end of the experiment. The standard deviation in freshwater was

initially constant at a value of ~0.05 mm, then at $t/t_c = 0.6$ it started to increase and reached a peak value of ~0.45 mm at $t/t_c = 0.87$. After that it decreased quickly to a minimum at $t/t_c = 1.23$ and then slowly increased.

In Fig. 12a the times series of frazil ice floc concentration, defined as the number of flocs per cm³, has a similar shape in the saline water experiments. The floc concentration increased from zero initially, reached a maximum value and then remained

approximately constant until the end of the experiments. In freshwater, the trend was slightly different with the floc concentration reaching a maximum value of 0.25 cm⁻³ at $t/t_c = 1.27$ and then decreasing slowly. In saline water the maximum floc concentrations were reached at approximately $t/t_c = 1.4$ and had values of approximately 1.10, 1.37, and 1.86 cm⁻³ at salinities of 15 ‰, 25 ‰, and 35 ‰, respectively. Figure 12b shows that as time progressed, the general trend for all cases is that the mean floc size continually increased with time. Initially (i.e. $t/t_c \leq 0.8$) the time series are noisy, particularly for the

freshwater case, because only a very small number of frazil ice flocs were produced during this time. At $t/t_c \geq 0.8$ the floc concentration has increased sufficiently that the noise disappears in all cases.  For the freshwater case, the mean floc size increased from 1.70 mm at $t/t_c \sim 1$ to 2.20 mm at $t/t_c \sim 1.13$ and continued increasing very slowly to 2.40 mm until $t/t_c = 2.0$. In saline water the mean floc size varied slightly with salinity and was always smaller than the freshwater floc sizes. In general, the mean floc size in saline water increased from ~0.5 mm at the start of the experiments up to ~1.90 mm at the end.

The average floc growth rate was 0.408, 0.118, 0.089, and 0.072 mm/min for the freshwater, 15 ‰, 25 ‰, and 35 ‰, respectively. Figure 12c shows that the time series of the standard deviation of floc size at $t/t_c \leq 1.0$ were quite noisy due to the small floc concentration. At $t/t_c \geq 1.0$ the standard deviation curves in all cases was approximately constant and varied between ~0.15 mm to ~0.25 mm with the lowest value being in freshwater.

### 7.2 Size and volume distributions

Figures 13 to 15 present the plots of the size distributions of frazil particles, flocs, and volume distributions of flocs for the 35 ‰ experiments during the three phases and the overall average (i.e. averaged over the entire duration). Although only distributions from the 35 ‰ experiments are presented, plots at the other salinities were very similar. It can be seen in Fig. 13 that the particle size distributions during the three phases and the overall average closely follow a lognormal distribution. The mean and standard deviation of particles sizes at all salinities are presented in Table 3. The mean size of frazil ice

particles averaged over the entire duration ($0 \leq t/t_c \leq 2$) defined as the overall mean size were 0.52, 0.46, 0.48 and 0.45 mm for salinities of  0 ‰,  15 ‰, 25 ‰ and 35 ‰, respectively. The overall mean particle size in freshwater was 8.3 % to 15.6 % larger than in saline water. The standard deviation decreased as salinity increased in all phases and overall. Also, the

standard deviation decreased with time for all salinities (from Phase 1 to Phase 3), with the largest decreases observed in the freshwater and the smallest decrease observed at 35 ‰. The data in Table 3 show that in saline water, the maximum mean particle size occurred during Phase 1 and decreased during the latter two phases 2 and 3. Conversely, for the freshwater experiments, the maximum mean particle size occurred during Phase 2.

All the floc size distributions plotted in Fig. 14 closely followed a lognormal distribution. A summary of the mean floc sizes can be found in Table 4. The overall mean floc sizes were 2.57, 1.64, 1.61, and 1.47 mm for salinities of 0 ‰, 15 ‰, 25 ‰ and 35 ‰, respectively. The mean size of the freshwater flocs was significantly larger than the saline water flocs, approximately 60% larger, and this is consistent with qualitative observations of the raw images. The mean size of saline
water flocs decreased slightly (i.e. 10 %) as salinity increased from 15 ‰ to 35 ‰. The standard deviation of floc sizes in freshwater was significant larger than in saline water and it decreased as salinity increased. At all four salinities, as the experiment progressed the mean size and standard deviation of the flocs increased and reached its maximum during Phase 3 of the experiment.

As discussed previously it was observed that there were very large flocs present in the freshwater experiments that were not observed in the saline water experiments. In order to assess this observation quantitatively, the 95th percentile of floc sizes, the maximum floc size and the mean size of flocs larger than the 95th percentile were computed, and these results are listed in Table 5. The 95th percentile of floc size was 6.91 mm in freshwater and decreased with increasing salinity to 3.96 mm at 35 ‰. The mean size of the largest 5 % of flocs varied from 11.9 mm in freshwater and decreased with increasing salinity to
5.38 mm at 35 ‰. The maximum floc size in freshwater was 95 mm compared to 23 to 36 mm in the saline water cases. The data in Table 5 supports the qualitative observation that the flocs were considerably larger in freshwater and that floc size decreased slightly as the salinity increased from 15 ‰ to 35 ‰.

Similar to the frazil ice floc size distributions, the frazil ice floc volume distributions during the three phases and the overall
distribution all fit a lognormal distribution closely as shown in Fig. 15. A summary of the estimated volume of ice in flocs can be found in Table 6. The mean estimated frazil floc volumes were 8.79, 1.14, 0.82, and 0.60 mm³ for freshwater, 15 ‰, 25 ‰, and 35 ‰, respectively. This equates to the average floc volume being 8 to 15 times greater in freshwater than in saline water.

## 7.3 Floc porosity and eccentricity

The volume concentration of ice at the end of the principal supercooling in freshwater, 15 ‰, 25 ‰, and 35 ‰ was estimated to be 0.0028, 0.0031, 0.0028, and 0.0021 m³/m³, respectively. These volume concentrations can be compared to the theoretical values calculated by considering the thermodynamic conditions of the experiment assuming the tank is fully





mixed (Osterkamp, 1978; Ye et al., 2004). The total heat exchanged with the surrounding environment, $Q_{tw}$ in W/m$^3$ is given by:

$$Q_{tw} = \rho C_p \frac{dT}{dt},$$

(3)

where, $\rho$ is the density of water in kg/m$^3$, $C_p$ is the specific heat of the water in J/kg.°C, and $dT/dt$ is the cooling rate in °C/s.

Density and specific heat depend on salinity and were calculated following Gill (1982). The cooling rate was obtained by calculating the slope of the supercooling curve over an interval beginning 15 min prior to freezing and ending at the freezing point. The volume concentration of frazil ice, $M_{sp}$ in m$^3$/m$^3$ is then given by:

$$M_{sp} = \frac{Q_{tw} t_{sp}}{L_i \rho_i},$$

(4)

where, $t_{sp}$ is the principal supercooling time in s, $L_i$ is the latent heat of fusion for ice in J/kg (334 J/kg), and $\rho_i$ is the density

of ice kg/m$^3$ (917 kg/m$^3$). The time of principal supercooling was obtained by averaging the supercooling curves for each salinity. Based on these calculations the computed volume concentration of ice formed during the principal supercooling were 0.0019, 0.0027, 0.0028, and 0.0027 m$^3$/m$^3$ for freshwater, 15 ‰, 25 ‰, and 35 ‰, respectively. This information was also used to estimate the porosity of frazil flocs, by equating the theoretical volume concentration of ice computed thermodynamically using Eq. (4) to the estimated total floc volume computed from the images. This calculation gives

estimated porosities for frazil ice flocs of 0.86, 0.82, 0.80, and 0.75 for freshwater, 15 ‰, 25 ‰, and 35 ‰, respectively.

The mean eccentricity of frazil flocs was 0.84, 0.82, 0.81, and 0.81 for freshwater, 15 ‰, 25 ‰, and 35 ‰, respectively. This corresponds to a ratio of major to minor axis length of 1.84, 1.75, 1.71, and 1.71 in freshwater, 15 ‰, 25 ‰, and 35 ‰, respectively. These eccentricity values indicate that the average shape of flocs in saline water did not vary significantly and

that the average shape of flocs was slightly more elongated in freshwater compared to saline water.

## 8 Discussion

The ensemble average time series of frazil ice particle concentration showed that, in general, the higher the salinity, the higher the particle concentration (Fig. 11a). As the salt is rejected by the ice, this can create a small pocket around the ice particle with a higher salinity and slightly lower freezing point (Rees Jones and Wells, 2015), thus inhibiting the flocculation

process. This would result in fewer particles adhering and sintering together to form flocs, leaving more particles in suspension, and producing the observed result of more individual particles being present at higher salinities. All of the time series of the particle concentration follow a similar pattern of initially increasing, reaching a maximum, and then decreasing with time. This pattern can be explained by considering particle production and the flocculation process. The flocculation rate, defined as the rate at which particles are sintered to other particles or flocs, is a function of the particle concentration.

This is because when there are more particles there is higher likelihood of particle to particle and particle to floc collisions which present opportunities for sintering. Prior to the time when the peaks occur in the time series, particle production





exceeds the flocculation rate but as time progresses a point is reached where particle production equals the rate of flocculation and the peak occurs at this time. This balance is reached because the flocculation rate is increasing as can be seen in the time series plots of flocs concentration in Fig. 12a. As time progresses both the particle production and the rate of flocculation reach asymptotic values when the water is at the residual temperature.

The time series of the mean particle sizes in Fig. 11b show that there is a decreasing trend in growth rate with increasing salinity, and on average the average particles growth rate in freshwater was approximately 4 times larger than in saline water. The growth rate depends on the rate at which the latent heat released by crystal growth is transported away from the crystal (Daly, 1994). During ice production in saline water, the supercooling is reduced and salt rejected by the growing crystal needs to diffuse away, which slows down the crystal growth (Rees Jones and Wells, 2018). Also the peak size was reached sooner in saline water ($t/t_c \approx 0.8$) as opposed to $t/t_c \approx 1.0$ in freshwater (see Table 3 and Fig. 11b). This may be due to the fact that the nucleation starts sooner in the saline water experiments (i.e. particles are being produced earlier than in freshwater), and particles do not start to appear until around $t/t_c = 0.8$ in freshwater compared to $t/t_c = 0.4$ in saline water (Fig. 11a). Therefore, the freshwater particles have less time to grow during Phase 1 and reach their maximum mean size later in the process during Phase 2.

The time series plots in Fig. 12a showed that throughout an entire experiment the higher the salinity, the higher the floc concentration. As salinity increases, the strength of particle to particle bonds decreases and therefore flocs remain smaller in size. This is because as flocs grow larger the shear forces that cause them to rupture also increase and therefore weaker internal bonds leads to smaller flocs and higher floc concentrations (Hanley and Tsang, 1984). Eventually, at $t/t_c \approx 1.4$, the floc concentration curves reach a plateau which suggests that particle to floc sintering becomes the dominant process as this process produces a net zero contribution to floc concentration. This is logical because at this point the particle production rate has decreased, resulting in low particle concentrations which reduces the probability of two particles colliding to near zero. However, the concentration of flocs is high, making particle floc collisions much more probable. Note that in freshwater, floc concentration increased, reached a peak and then slowly decreased starting from $t/t_c \approx 1.2$. This can be explained by the visual observation that flocs in freshwater tended to rise to the surface before the experiment was completed, whereas saline water flocs tended to remain suspended until the end of the experiment. The larger mean size of freshwater flocs increases their buoyancy, and thus increases their ability to overcome the turbulence of the flow and rise to the surface. This result is similar to observations made by Kempema et al. (1993).

The mean floc size increased as the experiments progressed (Fig. 12b) indicating that the mean volume of flocs also increased. As particle production continues, the probability of collisions of particles to flocs increases, resulting in a continuous increase of the mean floc size, especially in saline water. Later in the process, when the particle production slows down, most of the collisions would be particle to floc resulting in the slow steady growth of flocs until the end of the time of





the experiments. In freshwater, the rapid increase in floc sizes started at $t/t_c \approx 1$ with a rate that was 4 times faster than in the saline water but over a much shorter time interval.

The overall mean size of frazil ice particles in freshwater of 0.52 mm was slightly larger than in saline water. This value falls
within the reported ranges of previous studies of 23 μm to 5 mm (Gosink and Osterkamp, 1983; Daly and Colbeck, 1986; Ye et al., 2004; Clark and Doering, 2006; McFarlane et al., 2015 & 2017). The average overall mean particle sizes in saline water was 0.46 mm. This value is smaller than the previously reported estimates of particle size in saline water, which are on the order of 1 to 3 mm (Martin and Kauffman, 1981; Kempema et al., 1993; Ushio and Wakatsuchi, 1993; Smedsrud, 2001). However, these previously reported estimates of size ranges were based on very small sample sizes and using less
sophisticated measuring techniques including visual observations which may have influenced the results. The overall mean size of particles in freshwater were on average 1.13 times larger than particles in saline water. This suggests that the nucleation and growth of frazil ice particles are fairly similar in fresh and saline water. The particle size distributions at all salinities were lognormal, which is further evidence that the processes involved are similar.

The overall mean size of frazil ice flocs in freshwater was significantly larger than in saline water. Past studies have reported similar qualitative results that freshwater flocs are larger and more adhesive than saline water flocs (Hanley and Tsang, 1984; Kempema et al., 1993). The overall mean size of flocs in freshwater was 2.57 mm, which suggests that Clark and Doering (2009) measurements of flocs larger than 17 mm is not capturing the full spectrum of the floc size distribution. Flocs in freshwater were on average 1.64 times larger than in saline water. The maximum floc size observed in the
freshwater experiments was approximately three times larger than in the saline experiments. The difference in the flocculation process can be explained by freezing point depression in saline water. As the salt is rejected by the ice, it will increase the salinity of the surrounding water and slightly reduce the freezing point. This will inhibit the process of adhering and sintering, thus resulting in the mean size of frazil flocs decreasing with salinity.

The overall mean volume of flocs in freshwater was on average approximately 10 times larger than in saline water. It is interesting to note that there is large difference between the floc sizes when comparing freshwater to saline water, but the variation in saline water is less significant. This is further evidence that the flocculation process is largely dependent on whether the process occurs in saline water or freshwater, and less dependent on precise value of the salinity.

The volume concentrations estimated from the image processing are very close to the theoretical values computed thermodynamically. This indicates that assuming flocs are ellipsoid and have a porosity of 80% may provide a reasonable estimate of the ice volume concentration contained in flocs. Also, the deduced flocs porosities decreased with decreasing salinities from 0.86 for freshwater to 0.75 for at the highest salinity. These values are within the range of previously reported slush porosities (Beltaos, 2013; Ghobrial, 2012).





## 9 Conclusions

A total of 46 laboratory experiments were performed to determine the properties of individual frazil ice particles and flocs at salinities of 0 ‰, 15 ‰, 25 ‰, and 35 ‰. Visual examination of the images clearly showed that there were more irregular shaped particles in saline water than in freshwater. The average particle and floc growth rates decreased as salinity increased and the freshwater growth rates were ~ 4 times larger than the average growth rate in saline water. The mean frazil ice particle sizes ranged between 0.52 and 0.45 mm with particles sizes in freshwater ~ 13 % larger than in saline water. This indicates that mean particle sizes are only weakly dependent on salinity. The mean floc sizes ranged between 2.57 and 1.47 mm for freshwater and saline water with freshwater flocs being on average 60 % larger than in saline water. A lognormal distribution was observed to fit all the particle and flocs size distributions closely. The estimated flocs porosities were equal to 0.86, 0.82, 0.80, and 0.75 for salinities of 0 ‰, 15 ‰, 25 ‰, 35 ‰, respectively, which suggest that as salinity increases porosity decreases.

These laboratory measurements of the properties of frazil ice particles and flocs in saline water can be applied to oceanic frazil ice production models (Galton-Fenzi et al., 2012; Rees Jones and Wells, 2015 and 2018). Due to the complexity of the processes involved in the frazil ice production, all these models rely on the parameterization of small-scale phenomena and on the introduction of empirical constants (De Santi and Olla, 2017). Rees Jones and Wells (2018) noted that it is difficult to discriminate between models that use different crystal growth rate parameterizations because they can predict the same experimentally observed supercooling time series using different size distributions of initial seed crystals. They concluded that what was needed to discriminate between different models was for experiments to be conducted to measure crystal sizes as well as supercooling, which is precisely what we have done in this study. In a recent attempt to address the complexity of these processes, De Santi and Olla (2017) presented a simplified limit regime approach for ice formation based on the relative importance of salinity and the heat release due to the formation of frazil particles in controlling the supercooling. Therefore, results presented in this paper such as: the growth rate of frazil crystals, the size distributions during the different phases, and the corresponding supercooling curves can be used to evaluate crystal growth models adopted by earlier researcher (Svensson and Omstedt, 1994; Smedsrud and Jenkins, 2004; Holland et al., 2007; Galton-Fenzi et al., 2012; and Rees Jones and Wells, 2015). Additionally, in river ice models, inaccurate simulation of frazil dynamics can have significant implications on the freeze-up processes and overall development of the solid ice cover (Holland et al., 2007; Shen, 2010). The uncertainty in the prediction of the frazil nucleation, flocculation, and transport by the flow, as predicted by frazil ice dynamics models (e.g. Daly and Colbeck, 1986; Daly, 1994), results in inaccurate prediction of ice production rates. Ice volume concentration and deduced porosity results from the current study can be used to improve the accuracy of the estimates of ice discharges predicted by these models.





**Acknowledgements**

We would like to thank the Natural Sciences and Engineering Research Council of Canada (NSERC) for supporting this project (RGPIN-2015-04670 and RGPAS 477890-2015) and Perry Fedun for his valuable technical assistance. We are grateful for that support.

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



**Tables**

**Table 1: Summary of camera and polarizer set-ups used in the experiments.**

| Set-up # | Set-up 1 | Set-up 2 | Set-up 3 |
|---|---|---|---|
| **Experimental Conditions** | Particles in freshwater | Particles and flocs in saline water | Flocs in freshwater |
| **Polarizers Size (cm)** | 10 by 10 | 10 by 10 | 16 by 16 |
| **Space between Polarizers (cm)** | 2.2 | 2.2 | 3.5 |
| **ISO** | 6400 | 8000 | 6400 |
| **Shutter Speed (s)** | 1/2000 | 1/2000 | 1/2000 |
| **Aperture** | f/25 | f/29 | f/25 |
| **Image Frequency (Hz)** | 1 | 1 | 1 |
| **Camera Distance (cm)** | 5.7 | 8.7 | 30 |
| **Average Pixel size (µm)** | 6.4 | 8.3 | 28.8 |
| **Average field of view (width by height, mm)** | 47.5 by 31.7 | 61.3 by 40.9 | 162.9 by 141.3 |
| **Measuring Volume (cm³)** | 33.1 | 55.1 | 805.7 |
| **Lighting System** | Genaray SpectroLED | Genaray SpectroLED | Andoer FalconEyes |

**Table 2: Summary of statistics on the repeatability of experiments at each salinity showing the average μ, standard**
**deviation σ, and the coefficient of variation COV for the maximum supercooling and the cooling rate.**

| Salinity (‰) | Experimental Set-up # | # of repeated experiments | Maximum Supercooling (minimum temperature – freezing point) | | Cooling Rate | |
|---|---|---|---|---|---|---|
| | | | μ ± σ (°C) | COV (%) | μ ± σ (°C/min) | COV (%) |
| 0 | 1 | 10 | -0.0851 ± 0.0017 | 2.06 | 0.0123 ± 0.0004 | 2.89 |
| 15 | 2 | 9 | -0.0773 ± 0.0028 | 3.69 | 0.0112 ± 0.0003 | 2.87 |
| 25 | 2 | 9 | -0.0852 ± 0.0034 | 3.95 | 0.0095 ± 0.0003 | 3.02 |
| 35 | 2 | 9 | -0.0924 ± 0.0026 | 2.78 | 0.0094 ± 0.0007 | 7.11 |
| 0 | 3 | 9 | -0.0752 ± 0.0034 | 4.57 | 0.0092 ± 0.0003 | 3.73 |





**Table 3: Mean sizes and standard deviations of frazil ice particles during different phases and at all four salinities.**

| Salinity | Mean sizes ± standard deviations (mm) | | | Overall |
|---|---|---|---|---|
| | Phase 1 | Phase 2 | Phase 3 | |
| Freshwater | 0.54 ± 0.58 | 0.58 ± 0.46 | 0.48 ± 0.33 | 0.52 ± 0.41 |
| 15 ‰ | 0.54 ± 0.43 | 0.50 ± 0.36 | 0.42 ± 0.30 | 0.46 ± 0.35 |
| 25 ‰ | 0.53 ± 0.35 | 0.49 ± 0.34 | 0.44 ± 0.31 | 0.48 ± 0.33 |
| 35 ‰ | 0.50 ± 0.32 | 0.47 ± 0.32 | 0.42 ± 0.29 | 0.45 ± 0.31 |

**Table 4: Mean sizes and standard deviations of frazil ice flocs during different phases and at all four salinities.**

| Salinity | Mean sizes ± standard deviations (mm) | | | Overall |
|---|---|---|---|---|
| | Phase 1 | Phase 2 | Phase 3 | |
| Freshwater | 1.68 ± 1.19 | 2.28 ± 2.06 | 2.65 ± 3.09 | 2.57 ± 2.88 |
| 15 ‰ | 0.93 ± 0.96 | 1.45 ± 1.30 | 1.83 ± 1.81 | 1.64 ± 1.63 |
| 25 ‰ | 1.02 ± 0.81 | 1.39 ± 1.09 | 1.77 ± 1.57 | 1.61 ± 1.43 |
| 35 ‰ | 0.96 ± 0.82 | 1.30 ± 1.01 | 1.60 ± 1.40 | 1.47 ± 1.28 |

5  **Table 5: Sizes of the largest frazil ice flocs at all four salinities.**

| Salinity | 95th Percentile (mm) | Mean Size of Flocs Larger than 95th Percentile (mm) | Maximum Floc Size (mm) |
|---|---|---|---|
| Freshwater | 6.91 | 11.89 | 95.10 |
| 15 ‰ | 4.82 | 6.77 | 36.22 |
| 25 ‰ | 4.38 | 5.98 | 23.18 |
| 35 ‰ | 3.96 | 5.38 | 25.19 |




**Table 6: Estimated mean volumes and standard deviations of frazil ice flocs during different phases and at all four salinities.**

| Salinity | Mean volumes ± standard deviations (mm³) | | | Overall |
|---|---|---|---|---|
| | Phase 1 | Phase 2 | Phase 3 | |
| Freshwater | 0.40 ± 1.07 | 3.01 ± 37.68 | 10.67 ± 141.45 | 8.79 ± 117.98 |
| 15 ‰ | 0.21 ± 4.70 | 0.60 ± 4.36 | 1.54 ± 7.78 | 1.14 ± 6.68 |
| 25 ‰ | 0.15 ± 0.71 | 0.39 ± 1.49 | 1.08 ± 4.55 | 0.82 ± 3.78 |
| 35 ‰ | 0.16 ± 2.62 | 0.31 ± 1.05 | 0.78 ± 3.22 | 0.60 ± 2.72 |

**Figures**

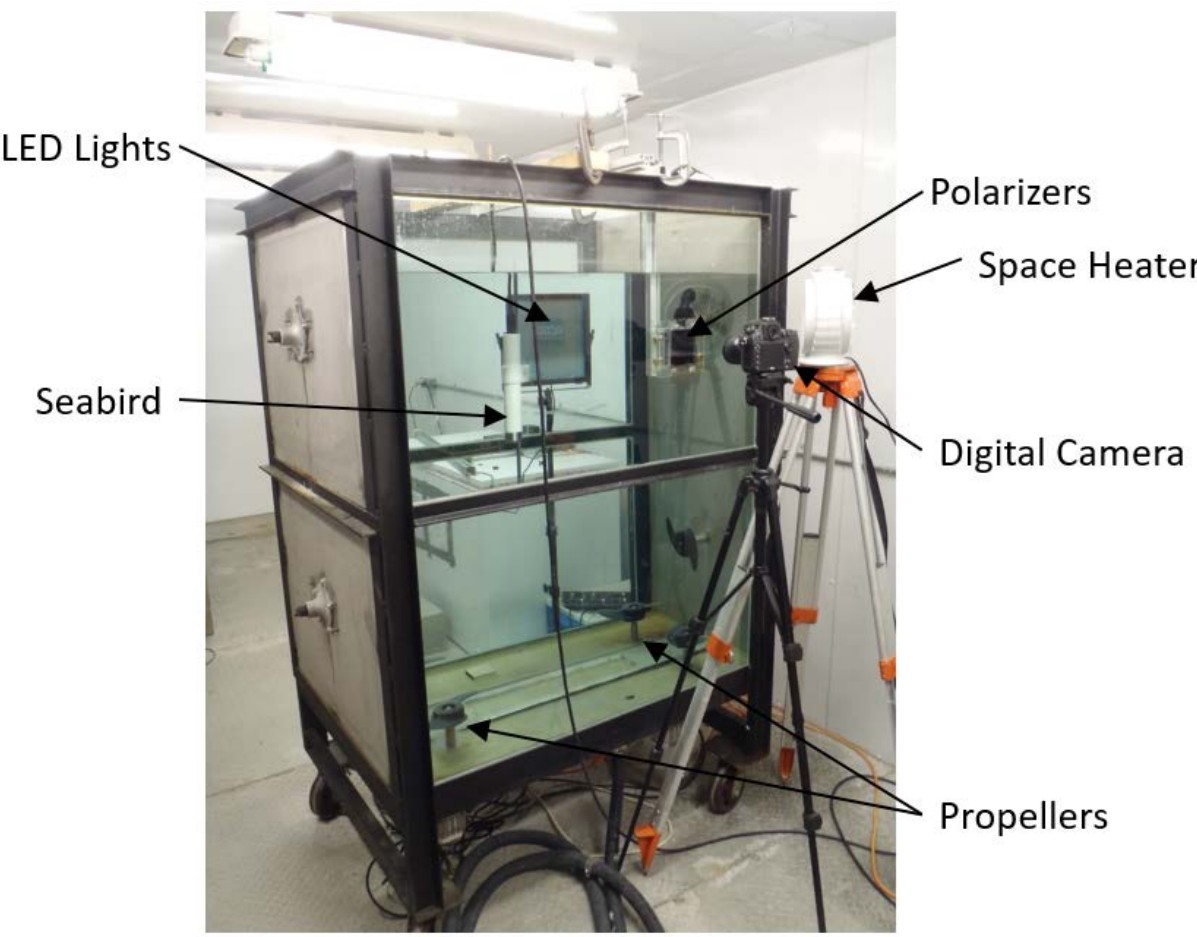

**Figure 1: Frazil ice tank experimental set-up.**



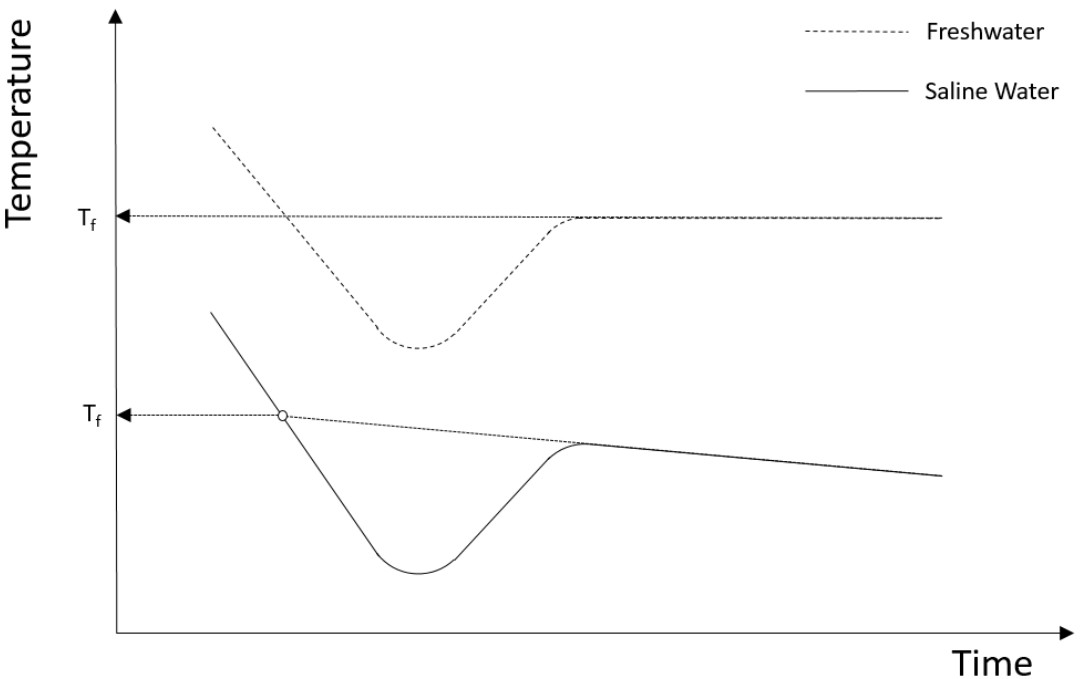

**Figure 2: Typical supercooling curves showing the difference between the freezing point temperature $T_f$ in freshwater and saline water. Adapted from She et al. (2016).**



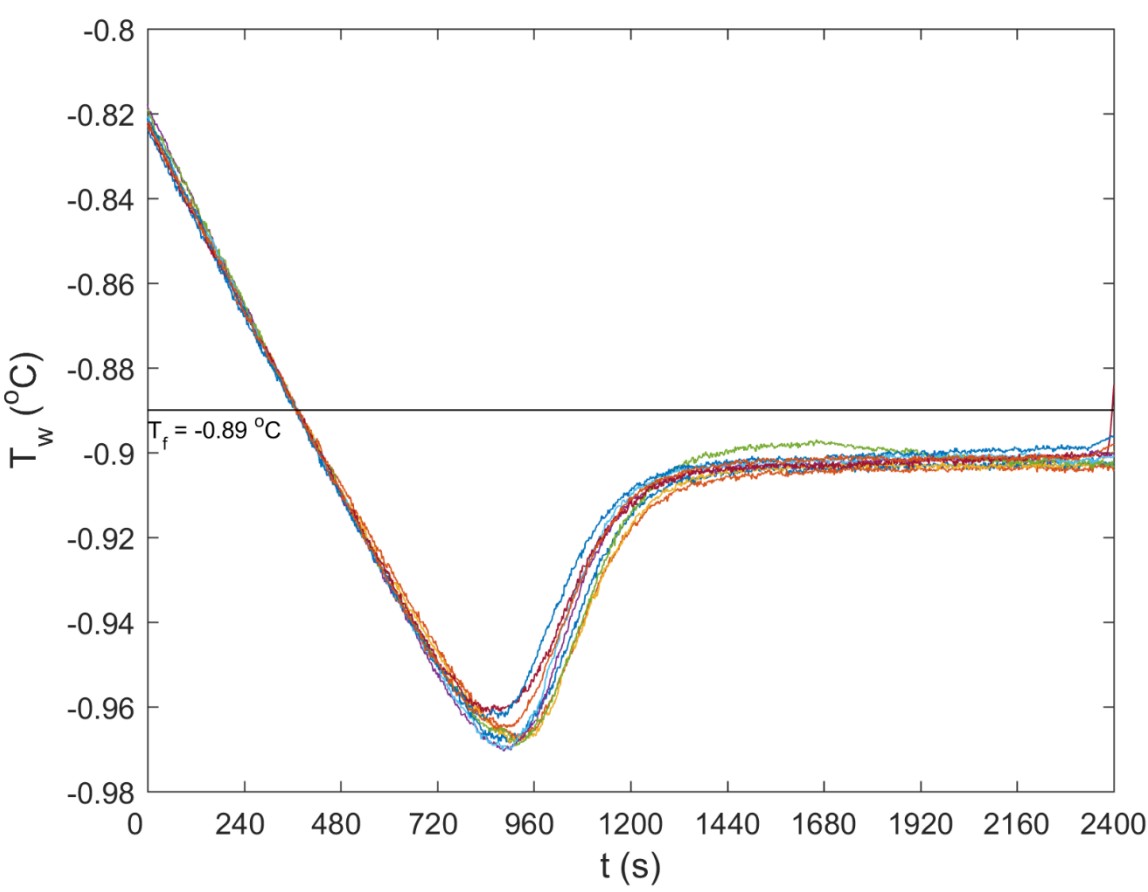

**Figure 3: Superimposed supercooling curves from repeated experiments showing water temperature, $T_w$ (°C), as a function of time, t (s), for a salinity of 15 ‰. The different curves represent the different experiments, and the freezing temperature $T_f$ (°C) is indicated by the horizontal line.**



(a)

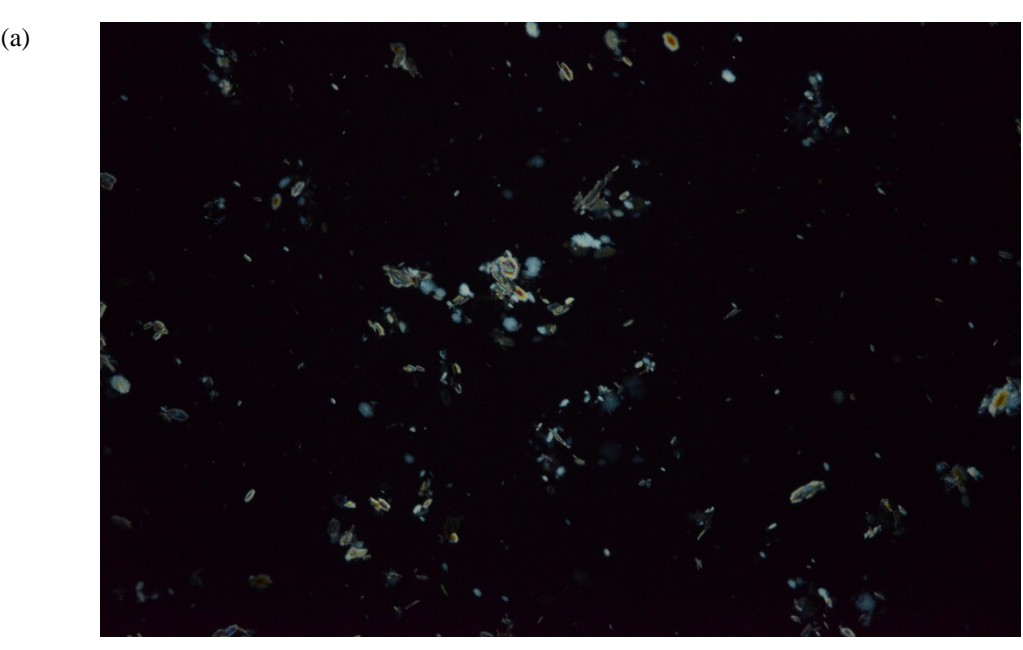

(b)

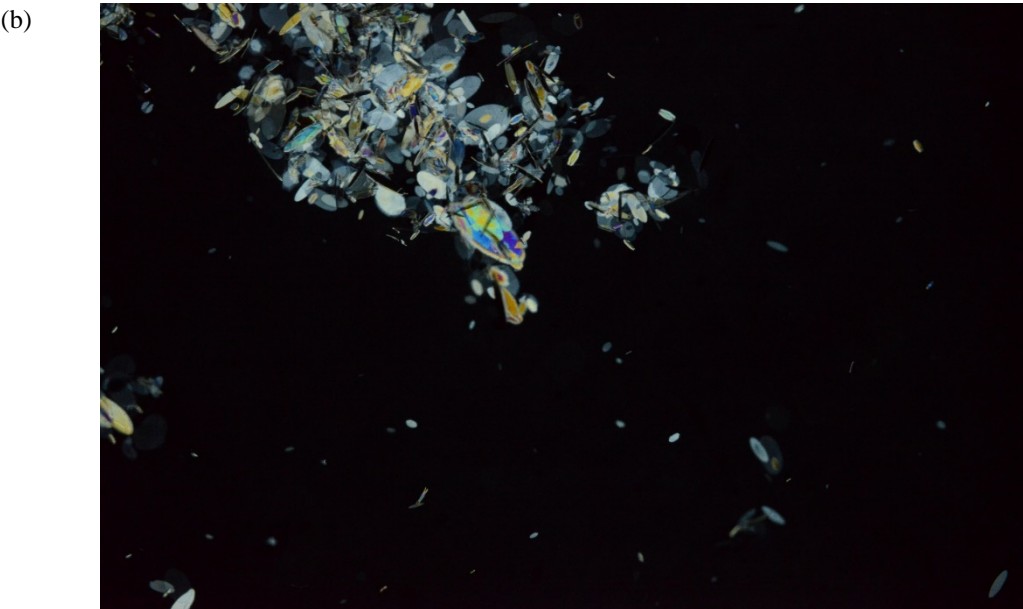

**Figure 4: Raw digital images showing frazil particles and flocs for a) 35 ‰ b) freshwater. Image dimensions are 3.07 cm by 4.61 cm.**

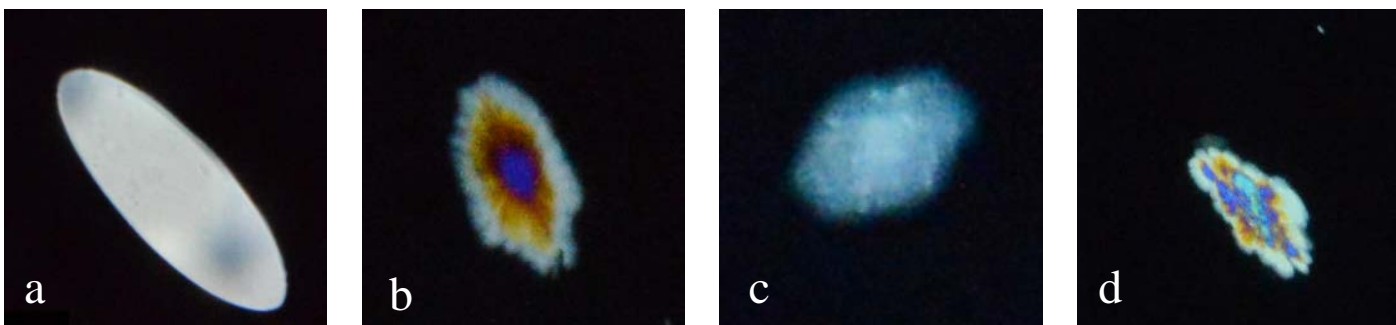

**Figure 5: Different shaped particles observed in the experiments: (a) disc-shaped, (b) dendritic, (c) hexagonal, and (d) irregular.**

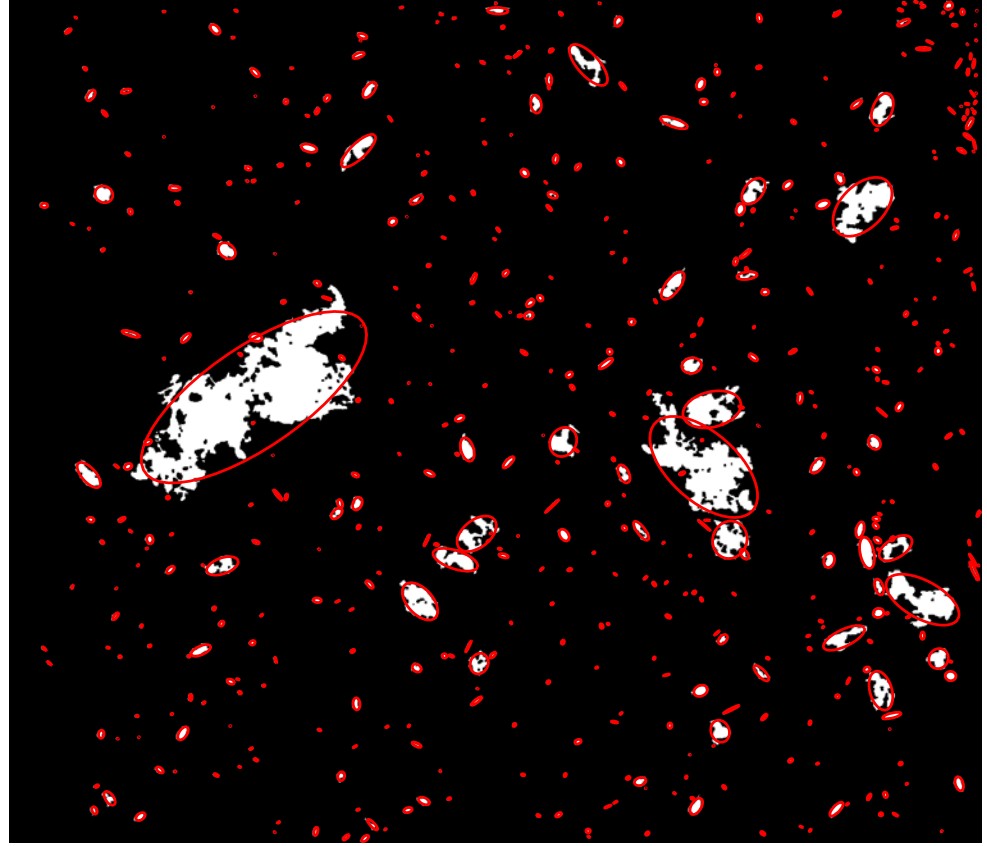

5    **Figure 6: Binary image with superimposed fitted ellipses plotted over each detected particle/floc.**



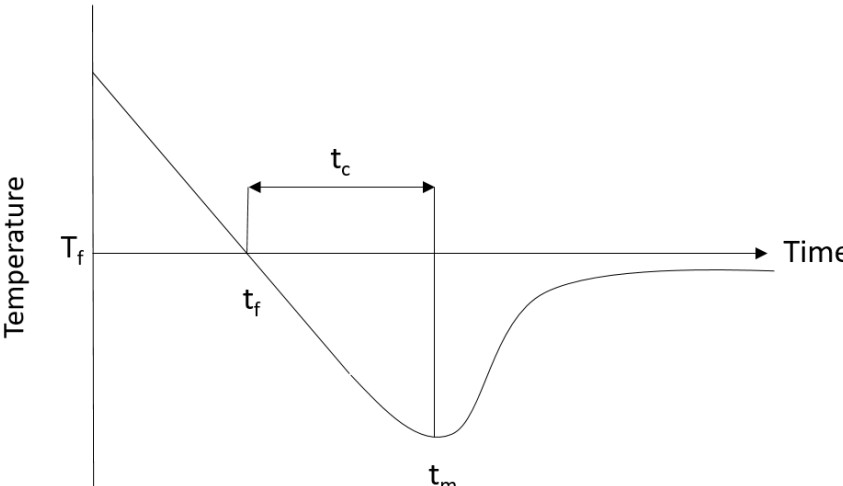

**Figure 7: Typical water supercooling curve as a function of time, where $t_f$ is the time of freezing, $t_m$ is the time of minimum temperature, $t_c$ is the time of cooling and $T_f$ is the freezing temperature.**


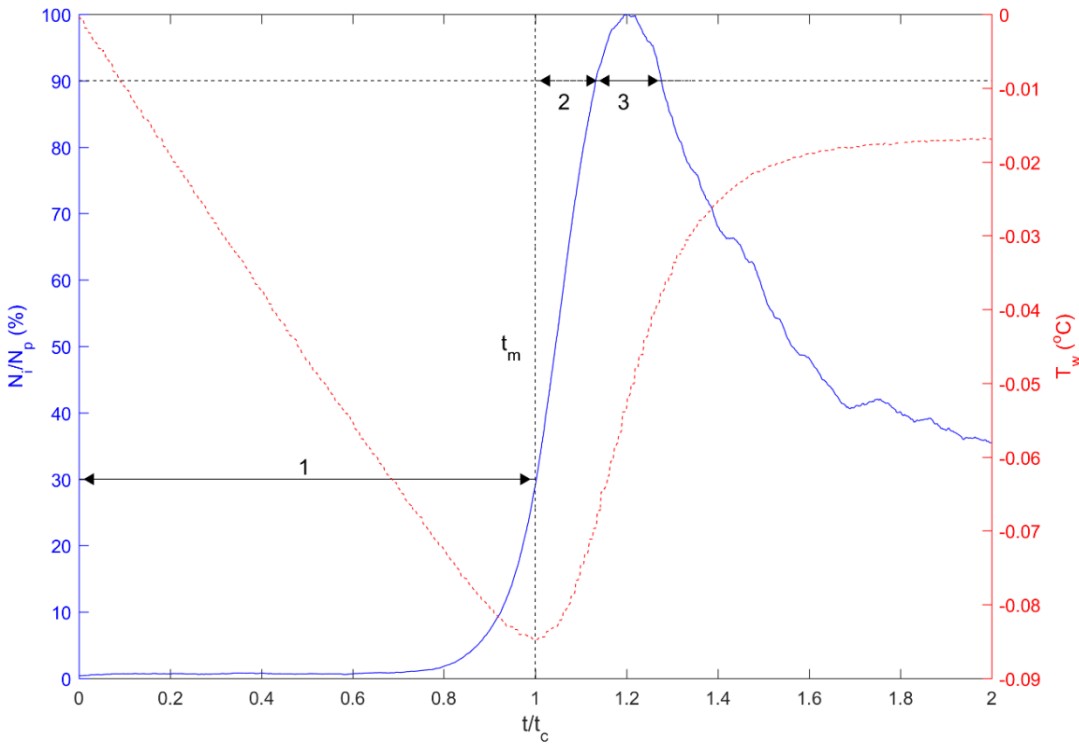

**Figure 8: Typical plot illustrating the three time-phases used to compare results from different salinities showing dimensionless number of particles or flocs per image (in blue) as a function of dimensionless time with superimposed supercooling curve (in red).** $N_i$ **is the number of particles per image,** $N_p$ **is the peak number of particles per image,** $T_w$
5 **is the water temperature (°C),** $t$ **is the time (s),** $t_c$ **is the time of cooling (s), and** $t_m$ **is the time of minimum temperature (s). Note: this plot is generated from a freshwater experiment where the freezing temperature** $T_f$ **is 0°C.**





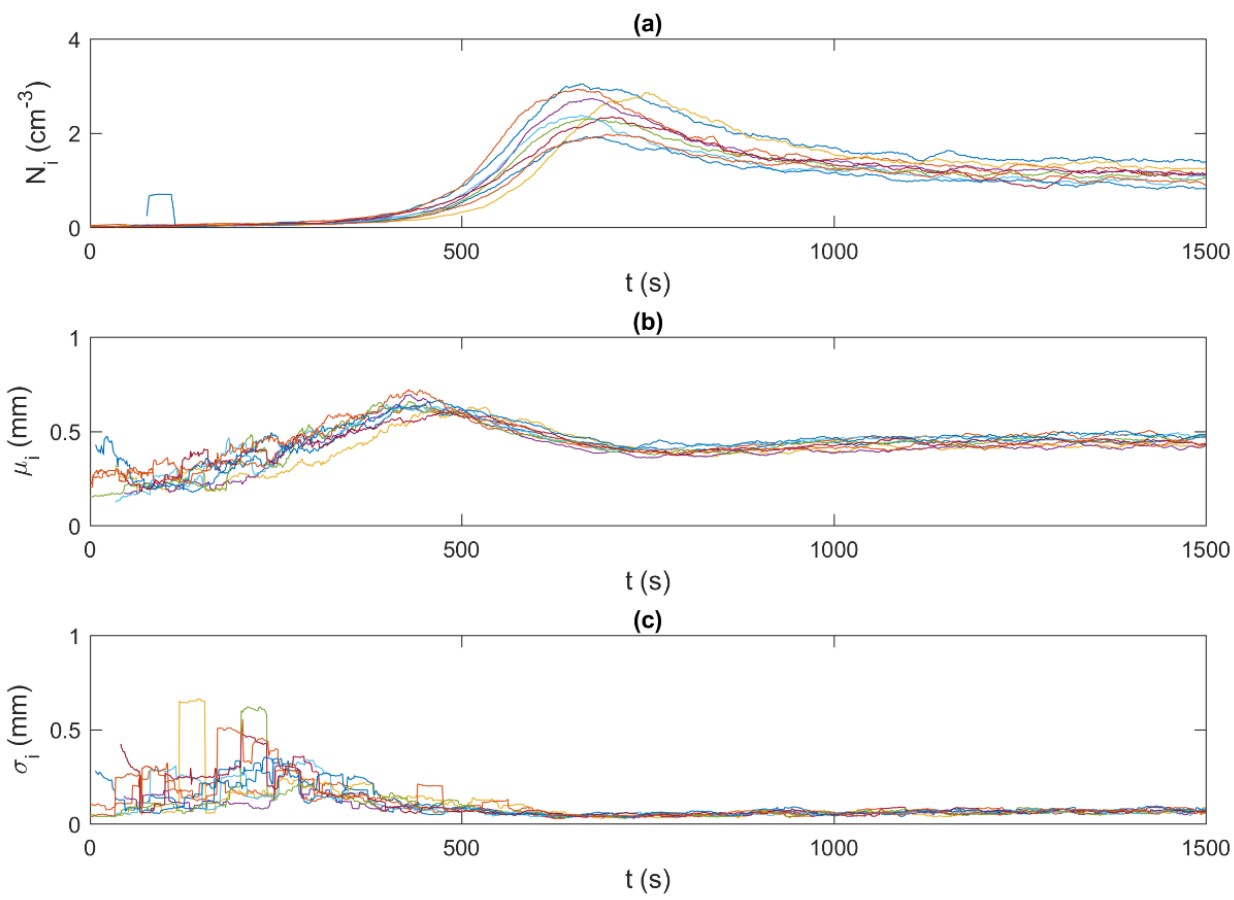

**Figure 9: Time series of the moving average frazil ice particle properties for all 15 ‰ experiments (shown in different colors). a) $N_i$, the frazil ice particle concentration (cm$^{-3}$), b) $\mu_i$, the mean size of frazil ice particles (mm), and c) $\sigma_i$, the standard deviation of the size of frazil ice particles (mm).**



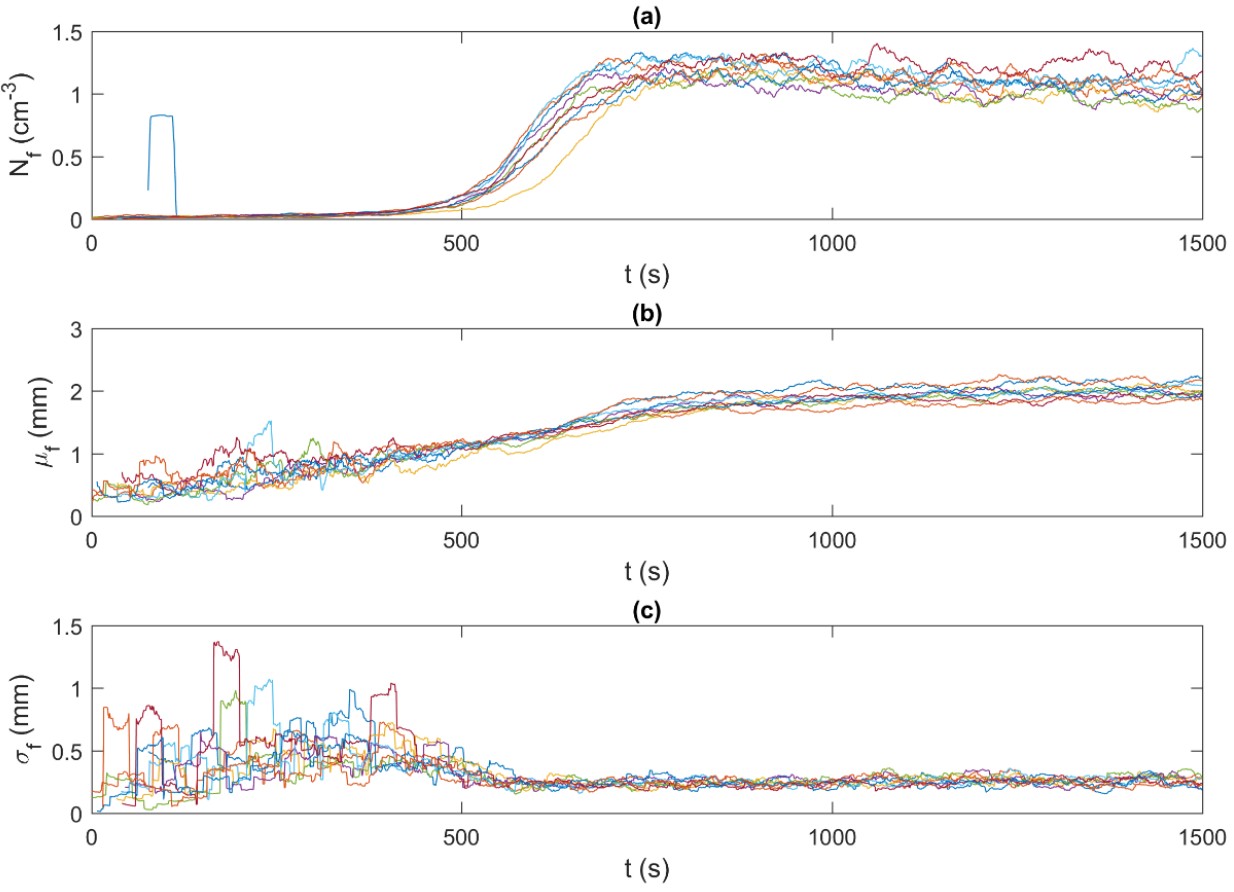

**Figure 10: Time series of the moving average frazil ice floc properties for all 15 ‰ experiments (shown in different colors). a) $N_f$, the frazil ice floc concentration (cm$^{-3}$), b) $\mu_f$, the mean size of frazil ice flocs (mm), and c) $\sigma_f$, the standard deviation of the size of frazil ice flocs (mm).**



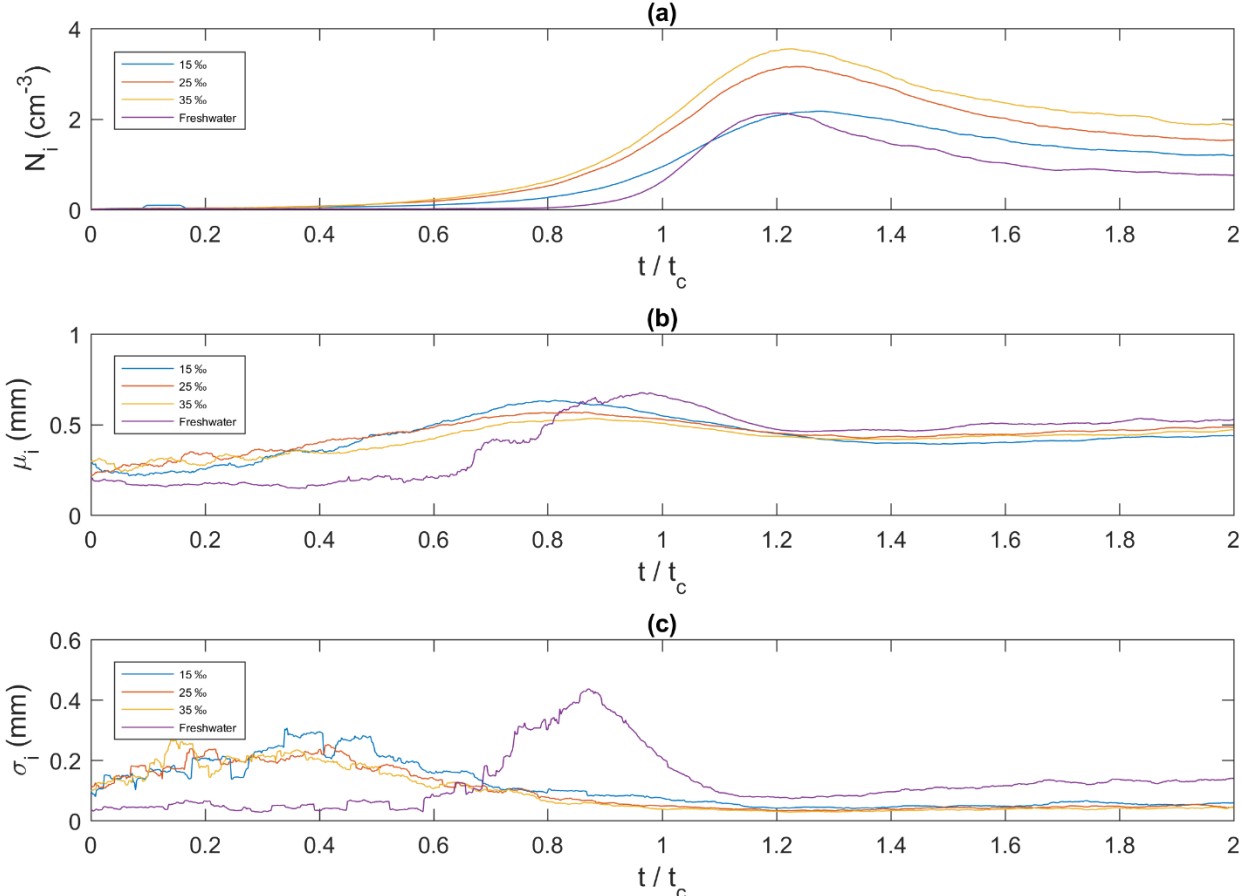

**Figure 11: Non-dimensional time series of the moving average frazil ice particle properties for all four salinities. a)** $N_i$, **the average frazil ice particle concentration (cm⁻³), b)** $\mu_i$, **the mean size of frazil ice particles (mm), and c)** $\sigma_i$, **the standard deviation of the size of frazil ice particles (mm).** $t/t_c = 1$ **corresponds to the time of the minimum temperature on the supercooling curves.**



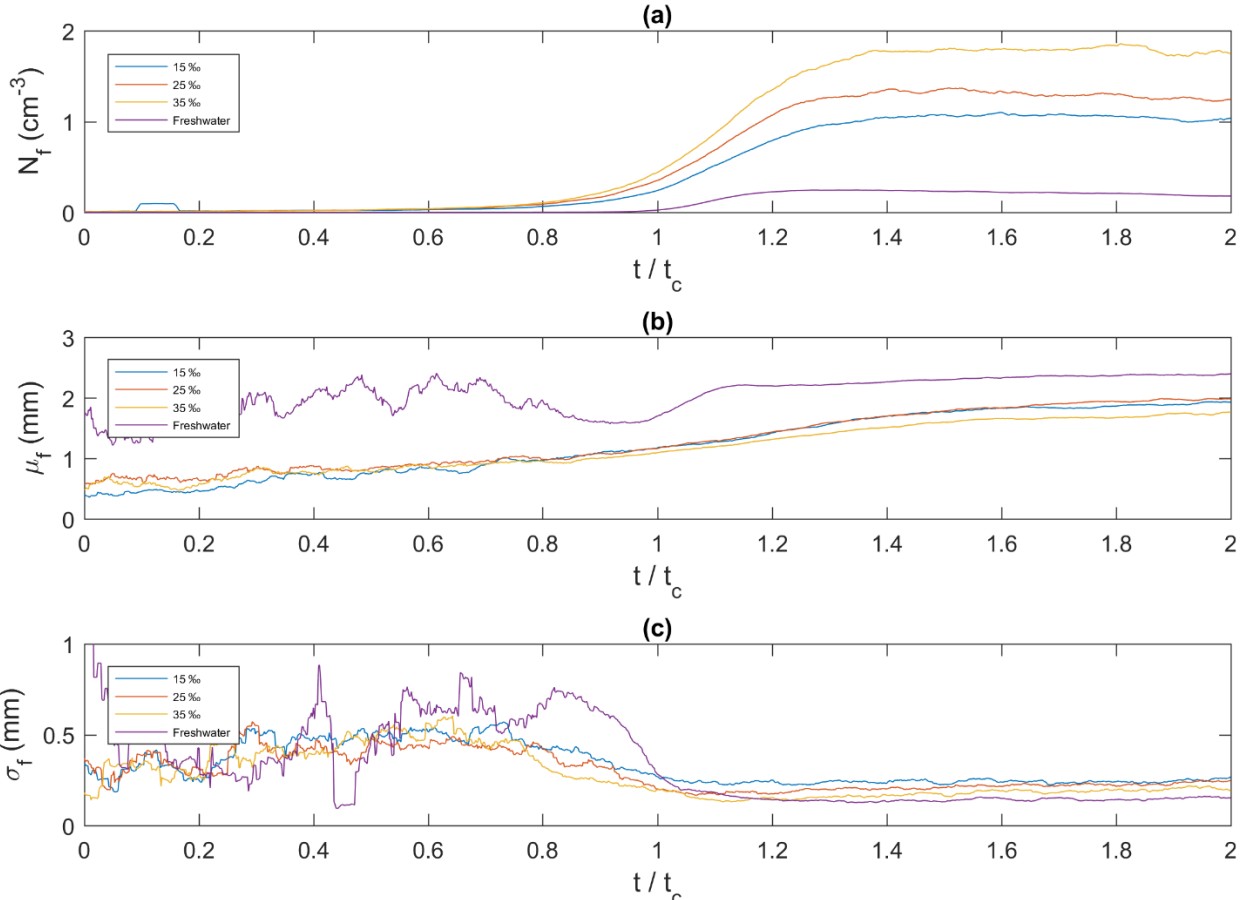

**Figure 12: Non-dimensional time series of the moving average frazil ice floc properties for all four salinities. a)** $N_f$**, the average frazil ice floc concentration (cm$^{-3}$) b)** $\mu_f$**, the mean size of frazil ice flocs (mm), and c)** $\sigma_f$**, the standard deviation of the size of frazil ice flocs (mm).** $t/t_c = 1$ **corresponds to the time of the minimum temperature on the supercooling curves.**



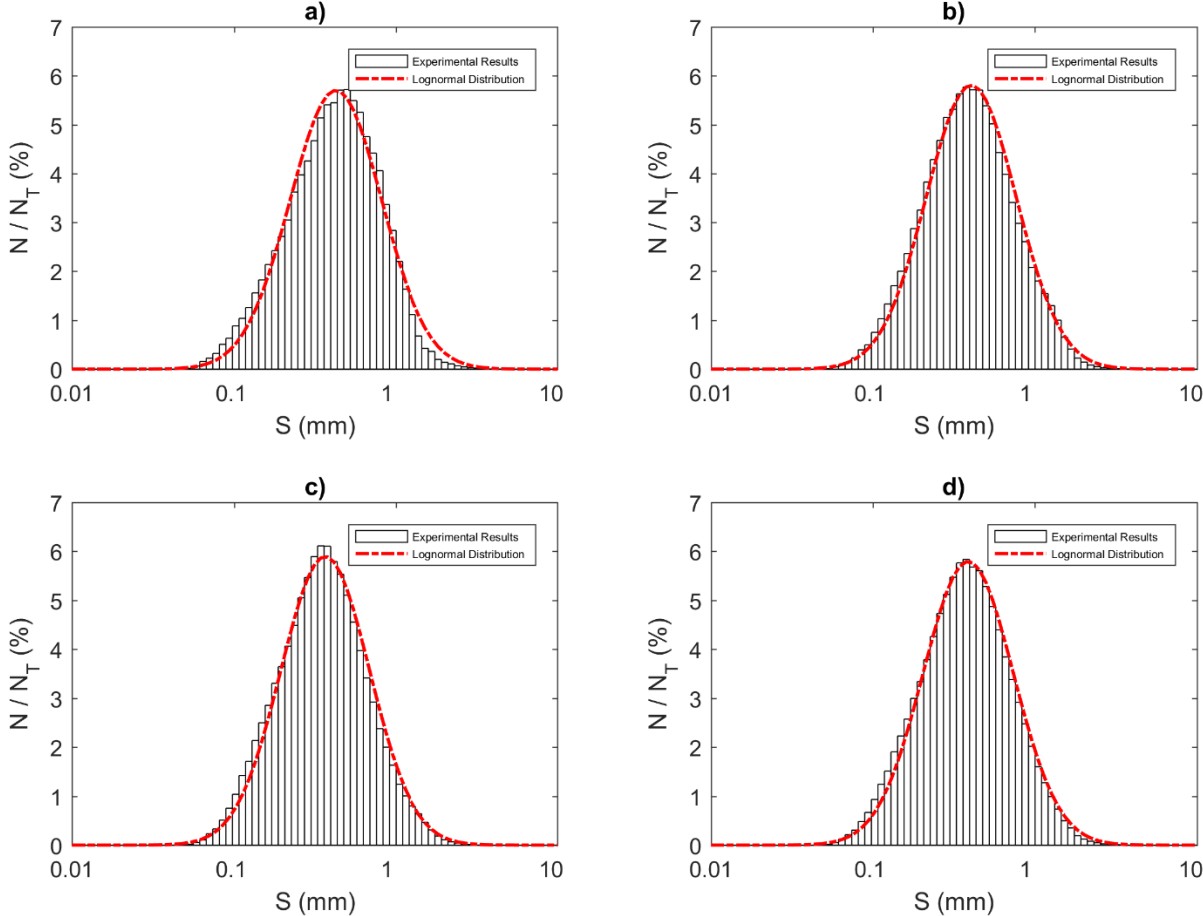

**Figure 13: Size distributions of individual particles at a salinity of 35 ‰ during a) Phase 1, b) Phase 2, c) Phase 3, and d) the entire time interval (i.e. averaged over all three phases). $N$ is the number of particles in each bin, $N_T$ is the total number of particles, and $S$ is the particle size (mm).**

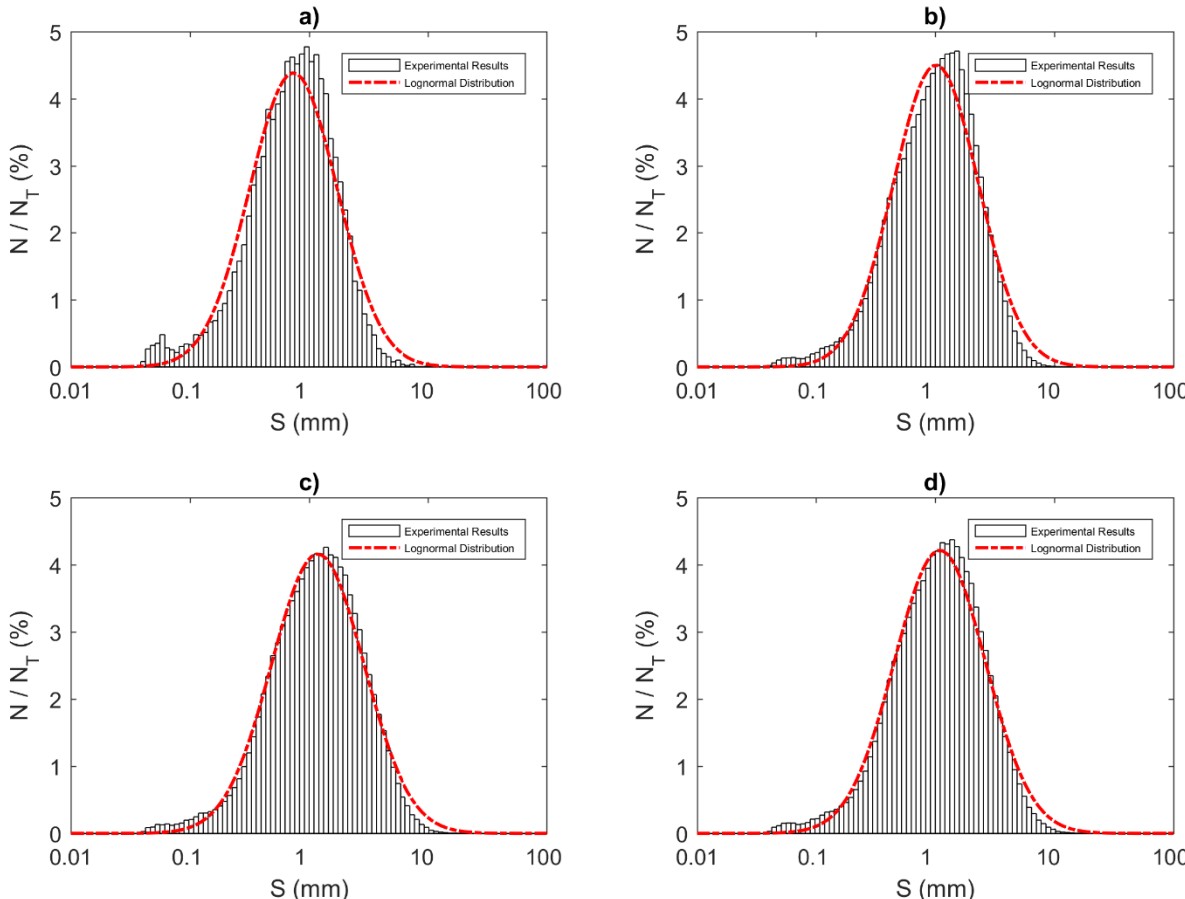

**Figure 14: Size distributions of frazil flocs at a salinity of 35 ‰ during a) Phase 1, b) Phase 2, c) Phase 3, and d) the entire time interval (i.e. averaged over all three phases).** $N$ **is the number of flocs in each bin,** $N_T$ **is the total number of flocs, and** $S$ **is the floc size (mm).**

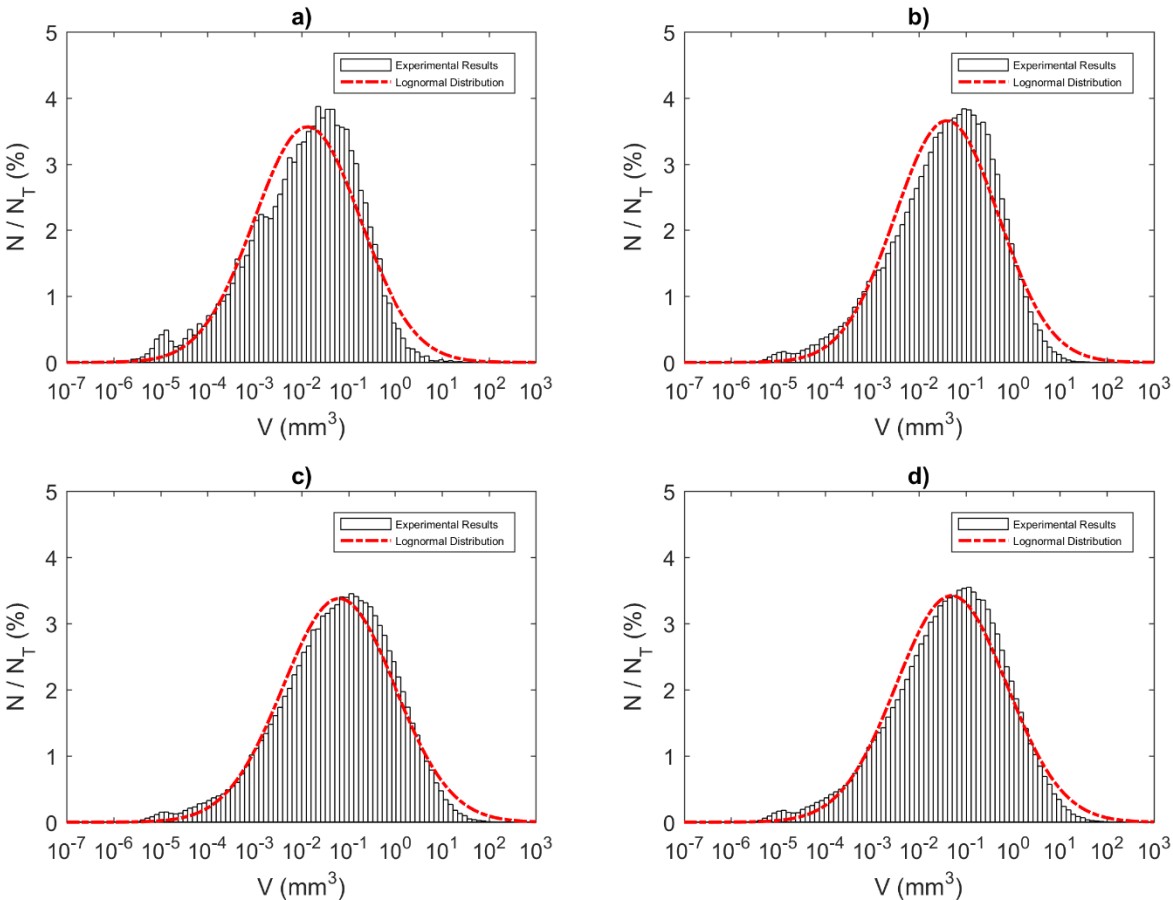

**Figure 15: Size distributions of frazil ice floc volume at a salinity of 35 ‰ during a) Phase 1, b) Phase 2, c) Phase 3, and d) the entire time interval (i.e. average over all three phases).** $N$ is the number of flocs in each bin, $N_T$ is the total number of flocs, and $V$ (mm$^3$) is the estimated volume of ice.

