# Peer review of "Laboratory Study of the Properties of Frazil Ice Particles and Flocs in Water of Different Salinities"

_The Cryosphere, 2019_

## Referee Comment (RC1) · Anonymous Referee #1 · 1 Jul 2019

The authors present laboratory experiments to measure frazil ice particles properties. Experiments are conducted in a water tank with bottom mounted propellers to create turbulence. The change in frazil ice properties as a function of salinity concentration is investigated, and different behavior between freshwater and saline water is highlighted. High-resolution camera and cross polarized lenses system is used to capture images and a suitable image processing algorithm is developed.

The growth rate of frazil crystals and flocs, their size distribution over time and the super cooling curves are measured and discussed. The presented findings suggest that overall process of nucleation and growth of frazil ice particles is similar for freshwater

and saline water: in both cases a lognormal particle size distribution is observed, even if in saline water the mean value of particle size is slightly smaller. On the contrary, flocculation process significantly slowdown in saline water. Furthermore, flocs porosity is estimated by comparison with a thermodynamic model.

Given the lack in measurements of the size and shape of frazil ice particles and flocs (particularly in saline water), the results of this paper can be very useful for modeler community. Moreover the authors deeply discuss the results with clear and precise comparison with literature data and models. Therefore I recommend this paper for publication.

I only have few comments for the authors.

- Turbulence intensity is held constant in all experiments with a turbulent kinetic energy dissipation rate of about 336 cmˆ2/sˆ3. Can the authors contextualize this value with those measured in ocean mixed layer or in rivers?

- In Introduction the rationale of this study is well presented and the state of art of the laboratory experiments is well detailed, but the novelty of the present study is quite hidden. I therefore suggest to improve this section (in particular to extend from line 6 to line 12 of page 4) by highlighting how the present study differs from previous ones.

- (very small comment) Page 1 line 23 I suggest to remove the "(i.e. cooled below 0°C)", since it is false for saline water.

---

## Referee Comment (RC2) · Anonymous Referee #2 · 7 Jul 2019

This paper presents a set of carefully executed laboratory experiments, measuring the number density and size distribution of individual frazil ice particles, and flocs of frazil crystals in waters of salinities varying from freshwater to sea water of 35 ppt. It provides new information in that it clearly demonstrates that a lognormal size distribution is observed in waters of all salinities. These are unique and carefully repeated measurements. The paper is very clearly written and is certainly worthy of publication.

I have two comments that would improve the paper, in my opinion. First, the short review of frazil production in rivers seems concise and complete. However in the ocean the authors only describe the production of frazil in polynyas. They cite Rees Jones &

[Figure]

Wells (2018) and Langhorne et al (2015) both of which are concerned with formation of frazil in a supercooled ice shelf water plume, yet there is no description of this process. The paper ought to briefly outline the process of frazil formation in ice shelf water as it differs from frazil formation due to heat loss to the atmosphere.

Second, measurements of temperature and supercooling are quoted to more significant figures than the accuracy of the measurements. This is unnecessary and misleading. Please consider rounding to the level of uncertainty of these and all derived quantities throughout the paper.

Technical Corrections p. 2, line 20 onwards: please include a description of frazil ice formation due to supercooling caused by pressure relief of upward-flowing ice shelf basal melt (e.g. see Langhorne et al (2015) and/or Rees Jones & Wells (2018) among many other references).

p. 2, Line 30: there is quite a large body of work on dense water formation and polynyas so it seems odd to mention one Arctic polynya from a rather old reference. The statement re dense water outflow is generally true, e.g. Ohshima et al. Global view of sea-ice production
in polynyas and its linkage to dense/bottom water formation,
 Geosci. Lett. (2016) 3:13 DOI 10.1186/s40562-016-0045-4

p. 4, line 6-12: as mentioned above, some processes of frazil formation under sea ice have not been discussed.

p. 4, line 21: how does turbulent kinetic energy dissipation in the laboratory tank compare with that in the ocean?

p. 4, line 27: change to "were used in experiments, either a 10 by 10 cm or a 16 by 16 cm polarizer". I tried to imagine how both were used at once.

p. 5, line 24: please round to a smaller number of significant figures to correctly reflect the uncertainty i.e. -0.003 to +0.005

p. 5, line 26: round to 0.0007 p. 6, line 11: please round to -8.0 $\pm$ 0.2

p. 7, line 6: please replace "exact freezing point" with "freezing point to better than 10 mK"

p. 7, line 8-9: please round to -0.89 ± 0.02, -1.48 ± 0.02 and -2.09 ± 0.02

p. 7, line 22: please consider significant figures in cooling rates.

p. 7, line 22: what is the COV?

p. 7, line 24-25: please consider rounding to 2 and 5%, and 3 and 7%

p. 8, line 15: "non-zero salinities"

p. 8, line 29-30: I didn't really understand the description of holes being filled as fig 6 clearly has holes.

p. 9, line 29-31: why should the diameter to thickness ratio of the floc be equal to that of the particle? Please can you discuss the expected error in c and hence in volume.

p. 12, line 29: arithmetic mean or geometric mean (which I believe is equal to the median)? For those not familiar with the lognormal distribution it might be useful to discuss the measures of the distribution (i.e. relationship of mean, median etc)

p. 13, line 26: mm3

p. 14, line 15 & p. 16, line 33: wow – fabulous. A porosity of 0.75 for 35 ppt agrees very well with estimates for frazil ice in layers under sea ice (called sub-ice platelet layer) (e.g. Langhorne et al, 2015).

p. 16, line 17-18: I'm not sure why the discrepancy between present measurements and those of Clark and Doering (2009) imply the latter are inadequate? Please explain.

Tables 2-5: Please reconsider rounding of all quantities. What is COV? Arithmetic mean or geometric mean (which I believe is equal to median)? For those not familiar with log normal distribution it might be useful to discuss the measures of the distribution (i.e. relationship of mean, median etc).

Fig 1: clearly not a "Seabird" ☺ Would it be better labelled "temperature sensor"?

Fig 2 caption: "saline water in a confined vessel" to account for the decrease in freezing point.

Figs 13-15: What is the value of NT in each figure? Mark the means on the distributions by vertical lines.

---

## Author Comment (AC1) · 10 Jul 2019

Authors Response to **Referee #1** (received and published: 1 July 2019)

The authors wish to thank Referee #1 for the constructive comments and corrections to the discussion paper. We have responded to each of the comments from the reviewer. The comments from the reviewer are in black font and our responses are in red font.

1. **Referee #1:**
   The authors present laboratory experiments to measure frazil ice particles properties. Experiments are conducted in a water tank with bottom mounted propellers to create turbulence. The change in frazil ice properties as a function of salinity concentration is investigated, and different behavior between freshwater and saline water is highlighted. High-resolution camera and cross polarized lenses system is used to capture images and a suitable image processing algorithm is developed.

   The growth rate of frazil crystals and flocs, their size distribution over time and the super cooling curves are measured and discussed. The presented findings suggest that overall process of nucleation and growth of frazil ice particles is similar for freshwater and saline water: in both cases a lognormal particle size distribution is observed, even if in saline water the mean value of particle size is slightly smaller. On the contrary, flocculation process significantly slowdown in saline water. Furthermore, flocs porosity is estimated by comparison with a thermodynamic model.

   Given the lack in measurements of the size and shape of frazil ice particles and flocs (particularly in saline water), the results of this paper can be very useful for modeler community. Moreover, the authors deeply discuss the results with clear and precise comparison with literature data and models. Therefore, I recommend this paper for publication.

   **Authors Response:**
   Thank you for your concise summary of our paper and for highlighting the significance of the presented results.

2. **Referee #1:**
   - Turbulence intensity is held constant in all experiments with a turbulent kinetic energy dissipation rate of about 336 cmˆ2/sˆ3. Can the authors contextualize this value with those measured in ocean mixed layer or in rivers?
   **Authors Response:**
   This is a very valid suggestion and it was also raised by the Referee #2. Although the dissipation rates in the tank were compared to the range of values estimated in rivers in Alberta (McFarlane et al., 2015), our initial submission did not compare this value to the reported ranges of dissipation rates in oceans. In general, the dissipation rates in oceans

range from $\sim 10^{-2}$ m$^2$/s$^3$ to $10^{-9}$ m$^2$/s$^3$ (Banner and Morrison, 2018; Wang and Liao, 2016) with a reported lower range in the Arctic regions ranging from $\sim 10^{-3}$ m$^2$/s$^3$ to $10^{-10}$ m$^2$/s$^3$ (Chanona et al., 2018; Scheifele et al., 2018). We will include a description of this limitation in the revised manuscript and will also point out the need for future experiments to investigate the behavior at very low dissipation rates.

3. **Referee #1:**
   - In Introduction the rationale of this study is well presented and the state of art of the laboratory experiments is well detailed, but the novelty of the present study is quite hidden. I therefore suggest to improve this section (in particular to extend from line 6 to line 12 of page 4) by highlighting how the present study differs from previous ones.
   **Authors Response:**
   We will expand on the last paragraph of the introduction to highlight the novelty of the current study. Specifically, we will highlight the fact that concurrent time series of supercooling temperatures with sizes and concentrations of particles and flocs observed at different salinities are being presented for the first time.

4. **Referee #1:**
   - (very small comment) Page 1 line 23 I suggest to remove the "(i.e. cooled below 0_C)", since it is false for saline water.
   **Authors Response:**
   Thank you for catching this mistake. The text will be updated as suggested.

---

## Author Comment (AC2) · 10 Jul 2019

Authors Response to **Referee #2** (received and published: 7 July 2019)

The authors wish to thank Referee #2 for the constructive comments and corrections to the discussion paper. We have responded to each of the comments from the reviewer. The comments from the reviewer are in black font while our responses are in red font.

1. **Referee #2:**
   This paper presents a set of carefully executed laboratory experiments, measuring the number density and size distribution of individual frazil ice particles, and flocs of frazil crystals in waters of salinities varying from freshwater to sea water of 35 ppt. It provides new information in that it clearly demonstrates that a lognormal size distribution is observed in waters of all salinities. These are unique and carefully repeated measurements. The paper is very clearly written and is certainly worthy of publication.
   **Authors Response:**
   Thank you for your positive comments and recommendation.

2. **Referee #2:**
   I have two comments that would improve the paper, in my opinion. First, the short review of frazil production in rivers seems concise and complete. However, in the ocean the authors only describe the production of frazil in polynyas. They cite Rees Jones & Wells (2018) and Langhorne et al (2015) both of which are concerned with formation of frazil in a supercooled ice shelf water plume, yet there is no description of this process. The paper ought to briefly outline the process of frazil formation in ice shelf water as it differs from frazil formation due to heat loss to the atmosphere.
   **Authors Response:**
   We agree that our initial submission should have discussed the process of frazil ice formation in supercooled ice shelf water plume. We will include a brief description of this process in the introduction similar to the description in Smedsrud and Jenkins (2003), Langhorne et al (2015), and Rees Jones & Wells (2018).

3. **Referee #2:**
   Second, measurements of temperature and supercooling are quoted to more significant figures than the accuracy of the measurements. This is unnecessary and misleading. Please consider rounding to the level of uncertainty of these and all derived quantities throughout the paper.
   **Authors Response:**
   We agree with this comment and the results in the paper will be updated so that the significant figures are consistent with the accuracy of the measurements.

4. **Referee #2:**
   Technical Corrections

p. 2, line 20 onwards: please include a description of frazil ice formation due to supercooling caused by pressure relief of upward-flowing ice shelf basal melt (e.g. see Langhorne et al (2015) and/or Rees Jones & Wells (2018) among many other references).
**Authors Response:**
Please see our response to comment number 2 above.

5. **Referee #2:**
p. 2, Line 30: there is quite a large body of work on dense water formation and polynyas so it seems odd to mention one Arctic polynya from a rather old reference. The statement re dense water outflow is generally true, e.g. Ohshima et al. Global view of sea-ice productionâˇ´lin polynyas and its linkage to dense/bottom water formation, âˇ´lGeosci. Lett. (2016) 3:13 DOI 10.1186/s40562-016-0045-4
**Authors Response:**
Thank you for providing the Ohshima et al (2016) reference. We will update the text to explain the general physics of the dense water outflow and reference Ohshima et al (2016) among others (e.g. Tamura et al., 2012; Nihashi and Ohshima, 2015)

6. **Referee #2:**
p. 4, line 6-12: as mentioned above, some processes of frazil formation under sea ice have not been discussed.
**Authors Response:**
Please see our response to comment number 2 above.

7. **Referee #2:**
p. 4, line 21: how does turbulent kinetic energy dissipation in the laboratory tank compare with that in the ocean?
**Authors Response:**
This issue was also raised by the Referee #1. Although the dissipation rates in the tank were compared to the range of values estimated in rivers in Alberta (McFarlane et al., 2015), our initial submission did not compare this value to the reported ranges of dissipation rates in oceans. In general, the dissipation rates in oceans range from ~ $10^{-2}$ $m^2/s^3$ to $10^{-9}$ $m^2/s^3$ (Banner and Morrison, 2018; Wang and Liao, 2016) with a reported lower range in the Arctic regions ranging from ~ $10^{-3}$ $m^2/s^3$ to $10^{-10}$ $m^2/s^3$ (Chanona et al., 2018; Scheifele et al., 2018). We will include a description of this limitation in the revised manuscript and will also point out the need for future experiments to investigate the behavior at very low dissipation rates.

8. **Referee #2:**
p. 4, line 27: change to "were used in experiments, either a 10 by 10 cm or a 16 by 16 cm polarizer". I tried to imagine how both were used at once.
**Authors Response:**
Updated.

9. **Referee #2:**
   p. 5, line 24: please round to a smaller number of significant figures to correctly reflect the uncertainty i.e. -0.003 to +0.005
   **Authors Response:**
   Updated.

10. **Referee #2:**
    p. 5, line 26: round to 0.0007
    **Authors Response:**
    Updated.

11. **Referee #2:**
    p. 6, line 11: please round to -8.0 ± 0.2
    **Authors Response:**
    Updated.

12. **Referee #2:**
    p. 7, line 6: please replace "exact freezing point" with "freezing point to better than 10 mK"
    **Authors Response:**
    Text updated to "this method yields values of the freezing point that are within 0.01 °C or better" (Mair et al, 1941; p. 610).

13. **Referee #2:**
    p. 7, line 8-9: please round to -0.89 ± 0.02, -1.48 ± 0.02 and -2.09 ± 0.02
    **Authors Response:**
    Updated.

14. **Referee #2:**
    p. 7, line 22: please consider significant figures in cooling rates.
    **Authors Response:**
    Updated.

15. **Referee #2:**
    p. 7, line 22: what is the COV?
    **Authors Response:**
    The COV stands for the coefficient of variation (standard deviation over the arithmetic mean). We updated the text to include this description. The COV was used as a measure of the repeatability of the experiments.

16. **Referee #2:**
    p. 7, line 24-25: please consider rounding to 2 and 5%, and 3 and 7%
    **Authors Response:**
    Updated.

17. **Referee #2:**
    p. 8, line 15: "non-zero salinities"
    **Authors Response:**
    Updated.

18. **Referee #2:**
    p. 8, line 29-30: I didn't really understand the description of holes being filled as fig 6 clearly has holes.
    **Authors Response:**
    We agree that this sentence was misleading as it implied that dilation and erosion would fill all holes in the fitted ellipse. The process of dilation and erosion of the binary images is done to smooth and fill any insignificant holes (by 5 pixels) that were generated due to the thresholding of the images. This is not to fill the gaps in the flocs but to smooth the outside edges of the individual particles. We will update the text to clarify this process.

19. **Referee #2:**
    p. 9, line 29-31: why should the diameter to thickness ratio of the floc be equal to that of the particle? Please can you discuss the expected error in c and hence in volume.
    **Authors Response:**
    The diameter to thickness ratio was only used for estimating the volume of the individual frazil ice particles (p. 9 lines 32-33). For flocs, we assumed that their shape was approximated by a fitted ellipsoid as defined by Eq. 2. The value of $c$ (the floc dimension perpendicular to the plane of the image) was assumed to be equal to the average of $a$ and $b$ (the major and minor axis of the fitted in-plane ellipse) but not greater than the distance between the polarizers when using Eq. 2 to estimate floc volume. Based on our visual observations of flocs, this seemed to be a reasonable assumption.

    Regarding the expected error in $c$ and hence the volume of the floc: The computed eccentricity (section 7.3) for the flocs at all salinities ranged between 0.81 and 0.84 with an average value of 0.82, which translates to $b \approx 0.6\ a$ and accordingly $c \approx 0.8\ a$ and $c \approx 1.3\ b$, when $c$ is assumed to be the average of $a$ and $b$. The average volume of a floc in this case is $V_{mean} = 4/3\ \pi\ (0.48\ a^3)$. Next, we considered the two extreme cases where $c$ is equal to either $a$ or $b$ (the major or minor axes). When $c$ equals $b$, the volume $V_{min} = 4/3\ \pi\ (0.36\ a^3)$, and when $c$ is equal $a$, then the volume $V_{max} = 4/3\ \pi\ (0.60\ a^3)$. Therefore, the expected error in estimating the flocs volume would be ±25%. If $c$ is less than $b$ or greater than $a$ then the error would increase and this likely does occur but we think the vast majority of flocs fall within the limits, that is, have $c$ values that fall between $a$ and $b$.

20. **Referee #2:**

p. 12, line 29: arithmetic mean or geometric mean (which I believe is equal to the median)? For those not familiar with the lognormal distribution it might be useful to discuss the measures of the distribution (i.e. relationship of mean, median etc)
**Authors Response:**
The reported averages μ are the arithmetic means and the corresponding standard deviation σ. The text has been updated to clarify that these are the values of the arithmetic means. We will also include a brief description of the lognormal distribution and the parameters that define it.

21. **Referee #2:**
p. 13, line 26: mm3
**Authors Response:**
The units of the volume are indeed $mm^3$.

22. **Referee #2:**
p. 14, line 15 & p. 16, line 33: wow – fabulous. A porosity of 0.75 for 35 ppt agrees very well with estimates for frazil ice in layers under sea ice (called sub-ice platelet layer) (e.g. Langhorne et al, 2015).
**Authors Response:**
Thank you for highlighting this. We will include in the discussion section the agreement between our estimates of flocs porosities and the values reported by Langhorne et al (2015).

23. **Referee #2:**
p. 16, line 17-18: I'm not sure why the discrepancy between present measurements and those of Clark and Doering (2009) imply the latter are inadequate? Please explain.
**Authors Response:**
Clark and Doering (2009) used a simplified criterion to identify individual flocs in the images, which was "a floc was considered to be any particle with an equivalent diameter greater than 17 mm" as quoted from their paper. Therefore, they disregarded any flocs smaller than this value and consequently they overestimated the mean sizes of flocs.

24. **Referee #2:**
Tables 2-5: Please reconsider rounding of all quantities. What is COV? Arithmetic mean or geometric mean (which I believe is equal to median)? For those not familiar with log normal distribution it might be useful to discuss the measures of the distribution (i.e. relationship of mean, median etc).
**Authors Response:**
Please see our response to comment number 3 and 20 above.

25. **Referee #2:**
Fig 1: clearly not a "Seabird". Would it be better labelled "temperature sensor"?
**Authors Response:**

Thank you for highlighting this, we updated the label to read "temperature sensor" as suggested.

26. **Referee #2:**
    Fig 2 caption: "saline water in a confined vessel" to account for the decrease in freezing point.
    **Authors Response:**
    Good point. The text has been updated.

27. **Referee #2:**
    Figs 13-15: What is the value of NT in each figure? Mark the means on the distributions by vertical lines.
    **Authors Response:**
    This is a good suggestion and we will add the values of NT and the arithmetic means to the captions for each figure.

---

## Referee Comment (RC3) · Anonymous Referee #3 · 24 Jul 2019

General Comments

The authors have conducted a very thorough analysis of the effect of various levels of salinity on frazil ice formation and flocculation. The manuscript is very well organized and written, and fits nicely within the scope of the journal. It fills a gap in the literature and will be well received by the river and sea ice researchers. I recommend that the paper be accepted.

The introduction provides the reader with a good appreciation of the current state of knowledge and clearly outlines the contribution of this paper. Throughout the paper the authors do a good job of comparing with previous experiments and field measurements

and highlighting similarities and differences.

The experimental setup is described well, and the authors have conducted an impressive 9 runs of each test condition to evaluate the reproducibility of the experiments. The data is presented well and the analysis is thorough. The conclusions are supported by the data. The entire study is tied up quite nicely by the end.

Specific Comments

Page 2, line 29. Am I correct that the authors are implying that this polynya is always there? I understand that polynya's are persistent, but is this one permanent? Should also be 'produces' rather than 'produce'.

Page 3, line 12 – might be nice to report the salinities of the Ushio and Wakatsuchi study rather than just saying low and high salinities.

Page 3, line 19 – Frazil 'deposits'. Line 22 – '44 ‰ water' should be reworded. Interchangeable use of disc and disk. As well as inconsistent hyphenation disc-shaped vs disc shaped, disk-like, etc.

If space is limited, Figure 2 could be removed. The experimentally-measured supercooling plot in Fig. 3 would be sufficient, and the 'theoretical' declining temperature of the residual supercooling level could just be described. COV is only defined in a figure caption.

Page 9, line 31 – should be Eq. (2) rather than 1.

The volume of particles being negligible compared to flocs is interesting, and a reader might question this based on the number of particles in Fig. 6 for example. Perhaps consider summing up the number and volume of particles and flocs in Fig. 6 and reporting them to help support your statement.

Page 11, line 10 – the eccentricity statements could appear near Eq. (1).

At a glance, Figure 9 and 10 show numbers and sizes of frazil particles that don't

appear to be much different than those of the frazil flocs. Have the size distributions plotted separately for Figs. 13 and 14 also make it difficult to compare. Consider combining these two figures.

Page 14, Line 9 – Latent heat of fusion of ice seems to be in J/g, rather than J/kg. The discussion sometimes switches from past to present tense in a way that doesn't seem to always be correct.
* * *

---

## Author Comment (AC3) · 27 Jul 2019

Authors Response to **Referee #3** (received and published: 24 July 2019)

The authors wish to thank Referee #3 for the constructive comments and corrections to the discussion paper. We have responded to each of the comments from the reviewer. The comments from the reviewer are in black font while our responses are in red font.

1. **Referee #3:**
   General Comments
   The authors have conducted a very thorough analysis of the effect of various levels of salinity on frazil ice formation and flocculation. The manuscript is very well organized and written, and fits nicely within the scope of the journal. It fills a gap in the literature and will be well received by the river and sea ice researchers. I recommend that the paper be accepted.

   The introduction provides the reader with a good appreciation of the current state of knowledge and clearly outlines the contribution of this paper. Throughout the paper the authors do a good job of comparing with previous experiments and field measurements. and highlighting similarities and differences.

   The experimental setup is described well, and the authors have conducted an impressive 9 runs of each test condition to evaluate the reproducibility of the experiments. The data is presented well and the analysis is thorough. The conclusions are supported by the data. The entire study is tied up quite nicely by the end.
   **Authors Response:**
   Thank you for your positive comments and recommendation.

2. **Referee #3:**
   Specific Comments
   Page 2, line 29. Am I correct that the authors are implying that this polynya is always there? I understand that polynya's are persistent, but is this one permanent? Should also be 'produces' rather than 'produce'.
   **Authors Response:**
   Thank you for highlighting this. The sentence now is reworded to read "In the polynya that forms in the Chukchi Sea off Cape Lisburne and Point Hope, Alaska, rapid ice growth occurs and is accompanied by an ocean salt flux that produces a dense outflow… etc". This will clarify that this polynya does form intermittently in this region.

3. **Referee #3:**
   Page 3, line 12 – might be nice to report the salinities of the Ushio and Wakatsuchi study rather than just saying low and high salinities.
   **Authors Response:**

We agree with this comment and the sentence has been reworded to read:" Ushio and Wakatsuchi (1993) performed laboratory experiments investigating the effect of wind conditions and salinity on the properties of frazil ice particles. In these experiments, the wind speed varied between 2 and 6 m/s, and water salinity between 0 and 35 ‰. They found that under strong wind and high salinity (above 25 ‰) conditions dendritic crystals having diameters of 2 to 3 mm were produced, and under calm air and lower salinity (below 25 ‰) conditions disc shaped crystals with diameters of approximately 5 mm were produced".

4. **Referee #3:**
Page 3, line 19 – Frazil 'deposits'. Line 22 – '44 ‰ water' should be reworded. Interchangeable use of disc and disk. As well as inconsistent hyphenation disc-shaped vs disc shaped, disk-like, etc.
**Authors Response:**
Page 3, line 19 now read:" Frazil deposits".
Line 22 now reads: "Hanley and Tsang (1984) performed laboratory experiments in a tank with water of 44 ‰ salinity and used a propeller to create turbulence".
Thank you for highlighting the inconsistency in these terms. Page 4, line 1 reads:" disc-like". Also, the manuscript has been reviewed to avoid this inconsistency.

5. **Referee #3:**
If space is limited, Figure 2 could be removed. The experimentally-measured supercooling plot in Fig. 3 would be sufficient, and the 'theoretical' declining temperature of the residual supercooling level could just be described. COV is only defined in a figure caption.
**Authors Response:**
Thank you for your suggestion. We will consider removing Figure 2 if we are out of space. The COV is now defined in the text in Page 7, line 22.

6. **Referee #3:**
Page 9, line 31 – should be Eq. (2) rather than 1.
**Authors Response:**
Updated.

7. **Referee #3:**
The volume of particles being negligible compared to flocs is interesting, and a reader might question this based on the number of particles in Fig. 6 for example. Perhaps consider summing up the number and volume of particles and flocs in Fig. 6 and reporting them to help support your statement.
**Authors Response:**
We agree that the fact that "the volume of particles is negligible compared to flocs" was not accurately addressed in the text. We've now updated the text on Page 10, line 2 to read:" In all salinities, it was found that the total volume of frazil ice particles during peak concentrations was between 0.3 and 1% of the total volume of frazil ice flocs.

Therefore, the volume of frazil ice particles was neglected when computing ice volumes".

8. **Referee #3:**
   Page 11, line 10 – the eccentricity statements could appear near Eq. (1).
   **Authors Response:**
   Updated, the sentence:" Ellipses have eccentricity between zero and one, where an eccentricity of zero corresponds to a circle and an eccentricity of one corresponds to a straight line." has been moved to Page 9 line 6.

9. **Referee #3:**
   At a glance, Figure 9 and 10 show numbers and sizes of frazil particles that don't appear to be much different than those of the frazil flocs. Have the size distributions plotted separately for Figs. 13 and 14 also make it difficult to compare. Consider combining these two figures.
   **Authors Response:**
   A careful examination at Figure 9a shows an average peak concentration of about 2 particle/cm$^3$, as opposed to ~ 1 floc/cm$^3$ from Figure 10a. We agree that Figures 13 and 14 might be difficult to compare especially since the x-axis is a log scale and has different limits in the two figures. We adjusted Figure 13 to have x-axis limits of 0.01 and 100 (mm) to match the limits in Figure 14 and this should facilitate comparison. We think this is a better solution than superimposing these two figures together since this would obscure the individual bars in each distribution and would make it difficult for the reader to distinguish between them.

10. **Referee #3:**
    Page 14, Line 9 – Latent heat of fusion of ice seems to be in J/g, rather than J/kg. The discussion sometimes switches from past to present tense in a way that doesn't seem to always be correct.
    **Authors Response:**
    Thank you for highlighting this. The value of the latent heat of fusion of ice is now updated to 3.33 x 10$^5$ J/kg. Also, the past tense has been used throughout this discussion for consistency.

---

## Author Response (AR1)

Authors Response to Referee #1 (received and published: 1 July 2019)

The authors wish to thank Referee #1 for the constructive comments and corrections to the discussion paper. We have responded to each of the comments from the reviewer. The comments from the reviewer are in black font and our responses are in red font.

**1. Referee #1:**

The authors present laboratory experiments to measure frazil ice particles properties. Experiments are conducted in a water tank with bottom mounted propellers to create turbulence. The change in frazil ice properties as a function of salinity concentration is investigated, and different behavior between freshwater and saline water is highlighted. High-resolution camera and cross polarized lenses system is used to capture images and a suitable image processing algorithm is developed.

The growth rate of frazil crystals and flocs, their size distribution over time and the super cooling curves are measured and discussed. The presented findings suggest that overall process of nucleation and growth of frazil ice particles is similar for freshwater and saline water: in both cases a lognormal particle size distribution is observed, even if in saline water the mean value of particle size is slightly smaller. On the contrary, flocculation process significantly slowdown in saline water. Furthermore, flocs porosity is estimated by comparison with a thermodynamic model.

Given the lack in measurements of the size and shape of frazil ice particles and flocs (particularly in saline water), the results of this paper can be very useful for modeler community. Moreover, the authors deeply discuss the results with clear and precise comparison with literature data and models. Therefore, I recommend this paper for publication.

**Authors Response:**

Thank you for your concise summary of our paper and for highlighting the significance of the presented results.

**2. Referee #1:**

- Turbulence intensity is held constant in all experiments with a turbulent kinetic energy dissipation rate of about 336 cm2/s3. Can the authors contextualize this value with those measured in ocean mixed layer or in rivers?

**Authors Response:**

This is a very valid suggestion and it was also raised by the Referee #2. Although the dissipation rates in the tank were compared to the range of values estimated in rivers in Alberta (McFarlane et al., 2015), our initial submission did not compare this value to the reported ranges of dissipation rates in oceans. In general, the dissipation rates in oceans

range from ~  $10^{-2} \text{ m}^2/\text{s}^3$  to  $10^{-9} \text{ m}^2/\text{s}^3$  (Banner and Morrison, 2018; Wang and Liao, 2016) with a reported lower range in the Arctic regions ranging from ~  $10^{-3} \text{ m}^2/\text{s}^3$  to  $10^{-10} \text{ m}^2/\text{s}^3$  (Chanona et al., 2018; Scheifele et al., 2018). We will include a description of this limitation in the revised manuscript and will also point out the need for future experiments to investigate the behavior at very low dissipation rates.

**3. Referee #1:**

- In Introduction the rationale of this study is well presented and the state of art of the laboratory experiments is well detailed, but the novelty of the present study is quite hidden. I therefore suggest to improve this section (in particular to extend from line 6 to line 12 of page 4) by highlighting how the present study differs from previous ones. **Authors Response:**

We will expand on the last paragraph of the introduction to highlight the novelty of the current study. Specifically, we will highlight the fact that concurrent time series of supercooling temperatures with sizes and concentrations of particles and flocs observed at different salinities are being presented for the first time.

**4. Referee #1:**

- (very small comment) Page 1 line 23 I suggest to remove the "(i.e. cooled below 0\_C)", since it is false for saline water.

**Authors Response:**

Thank you for catching this mistake. The text will be updated as suggested.
The authors wish to thank Referee #2 for the constructive comments and corrections to the discussion paper. We have responded to each of the comments from the reviewer. The comments from the reviewer are in black font while our responses are in red font.

**1. Referee #2:**

This paper presents a set of carefully executed laboratory experiments, measuring the number density and size distribution of individual frazil ice particles, and flocs of frazil crystals in waters of salinities varying from freshwater to sea water of 35 ppt. It provides new information in that it clearly demonstrates that a lognormal size distribution is observed in waters of all salinities. These are unique and carefully repeated measurements. The paper is very clearly written and is certainly worthy of publication. **Authors Response:**

Thank you for your positive comments and recommendation.

**2. Referee #2:**

I have two comments that would improve the paper, in my opinion. First, the short review of frazil production in rivers seems concise and complete. However, in the ocean the authors only describe the production of frazil in polynyas. They cite Rees Jones & Wells (2018) and Langhorne et al (2015) both of which are concerned with formation of frazil in a supercooled ice shelf water plume, yet there is no description of this process. The paper ought to briefly outline the process of frazil formation in ice shelf water as it differs from frazil formation due to heat loss to the atmosphere.

**Authors Response:**

We agree that our initial submission should have discussed the process of frazil ice formation in supercooled ice shelf water plume. We will include a brief description of this process in the introduction similar to the description in Smedsrud and Jenkins (2003), Langhorne et al (2015), and Rees Jones & Wells (2018).

**3. Referee #2:**

Second, measurements of temperature and supercooling are quoted to more significant figures than the accuracy of the measurements. This is unnecessary and misleading. Please consider rounding to the level of uncertainty of these and all derived quantities throughout the paper.

**Authors Response:**

We agree with this comment and the results in the paper will be updated so that the significant figures are consistent with the accuracy of the measurements.

**4. Referee #2:**

**Technical Corrections**

p. 2, line 20 onwards: please include a description of frazil ice formation due to supercooling caused by pressure relief of upward-flowing ice shelf basal melt (e.g. see Langhorne et al (2015) and/or Rees Jones & Wells (2018) among many other references).

**Authors Response:**

Please see our response to comment number 2 above.

**5. Referee #2:**

p. 2, Line 30: there is quite a large body of work on dense water formation and polynyas so it seems odd to mention one Arctic polynya from a rather old reference. The statement re dense water outflow is generally true, e.g. Ohshima et al. Global view of sea-ice productionâA~/lin polynyas and its linkage to dense/bottom water formation, âA~/lGeosci. Lett. (2016) 3:13 DOI 10.1186/s40562-016-0045-4

**Authors Response:**

Thank you for providing the Ohshima et al (2016) reference. We will update the text to explain the general physics of the dense water outflow and reference Ohshima et al (2016) among others (e.g. Tamura et al., 2012; Nihashi and Ohshima, 2015)

**6. Referee #2:**

p. 4, line 6-12: as mentioned above, some processes of frazil formation under sea ice have not been discussed.

**Authors Response:**

Please see our response to comment number 2 above.

**7. Referee #2:**

p. 4, line 21: how does turbulent kinetic energy dissipation in the laboratory tank compare with that in the ocean?

**Authors Response:**

This issue was also raised by the Referee #1. Although the dissipation rates in the tank were compared to the range of values estimated in rivers in Alberta (McFarlane et al., 2015), our initial submission did not compare this value to the reported ranges of dissipation rates in oceans. In general, the dissipation rates in oceans range from ~  $10^{-2}$  m2/s3 to  $10^{-9}$  m2/s3 (Banner and Morrison, 2018; Wang and Liao, 2016) with a reported lower range in the Arctic regions ranging from ~  $10^{-3}$  m2/s3 to  $10^{-10}$  m2/s3 (Chanona et al., 2018; Scheifele et al., 2018). We will include a description of this limitation in the revised manuscript and will also point out the need for future experiments to investigate the behavior at very low dissipation rates.

**8. Referee #2:**

p. 4, line 27: change to "were used in experiments, either a 10 by 10 cm or a 16 by 16 cm polarizer". I tried to imagine how both were used at once.

**Authors Response:**

Updated.

**9. Referee #2:**

p. 5, line 24: please round to a smaller number of significant figures to correctly reflect the uncertainty i.e. -0.003 to +0.005

Authors Response:

Updated.

**10. Referee #2:**

p. 5, line 26: round to 0.0007 Authors Response: Updated.

**11. Referee #2:**

p. 6, line 11: please round to -8.0 ± 0.2
 Authors Response:
 Updated.

**12. Referee #2:**

p. 7, line 6: please replace "exact freezing point" with "freezing point to better than 10 mK"

**Authors Response:**

Text updated to "this method yields values of the freezing point that are within 0.01 °C or better" (Mair et al, 1941; p. 610).

**13. Referee #2:**

p. 7, line 8-9: please round to  $-0.89 \pm 0.02$ ,  $-1.48 \pm 0.02$  and  $-2.09 \pm 0.02$ Authors Response: Updated.

**14. Referee #2:**

p. 7, line 22: please consider significant figures in cooling rates.
 Authors Response:
 Updated.

**15. Referee #2:**

p. 7, line 22: what is the COV?

**Authors Response:**

The COV stands for the coefficient of variation (standard deviation over the arithmetic mean). We updated the text to include this description. The COV was used as a measure of the repeatability of the experiments.

**16. Referee #2:**

p. 7, line 24-25: please consider rounding to 2 and 5%, and 3 and 7% **Authors Response:** Updated.

**17. Referee #2:**

p. 8, line 15: "non-zero salinities" Authors Response: Updated.

**18. Referee #2:**

p. 8, line 29-30: I didn't really understand the description of holes being filled as fig 6 clearly has holes.

**Authors Response:**

We agree that this sentence was misleading as it implied that dilation and erosion would fill all holes in the fitted ellipse. The process of dilation and erosion of the binary images is done to smooth and fill any insignificant holes (by 5 pixels) that were generated due to the thresholding of the images. This is not to fill the gaps in the flocs but to smooth the outside edges of the individual particles. We will update the text to clarify this process.

**19. Referee #2:**

p. 9, line 29-31: why should the diameter to thickness ratio of the floc be equal to that of the particle? Please can you discuss the expected error in c and hence in volume. **Authors Response:**

The diameter to thickness ratio was only used for estimating the volume of the individual frazil ice particles (p. 9 lines 32-33). For flocs, we assumed that their shape was approximated by a fitted ellipsoid as defined by Eq. 2. The value of *c* (the floc dimension perpendicular to the plane of the image) was assumed to be equal to the average of *a* and *b* (the major and minor axis of the fitted in-plane ellipse) but not greater than the distance between the polarizers when using Eq. 2 to estimate floc volume. Based on our visual observations of flocs, this seemed to be a reasonable assumption.

Regarding the expected error in *c* and hence the volume of the floc: The computed eccentricity (section 7.3) for the flocs at all salinities ranged between 0.81 and 0.84 with an average value of 0.82, which translates to  $b \approx 0.6 a$  and accordingly  $c \approx 0.8 a$  and  $c \approx 1.3 b$ , when *c* is assumed to be the average of *a* and *b*. The average volume of a floc in this case is  $V_{mean} = 4/3 \pi (0.48 a^3)$ . Next, we considered the two extreme cases where *c* is equal to either *a* or *b* (the major or minor axes). When *c* equals *b*, the volume  $V_{min} = 4/3 \pi (0.36 a^3)$ , and when *c* is equal *a*, then the volume  $V_{max} = 4/3 \pi (0.60 a^3)$ . Therefore, the expected error in estimating the flocs volume would be ±25%. If *c* is less than *b* or greater than *a* then the error would increase and this likely does occur but we think the vast majority of flocs fall within the limits, that is, have *c* values that fall between *a* and *b*.

**20. Referee #2:**

p. 12, line 29: arithmetic mean or geometric mean (which I believe is equal to the median)? For those not familiar with the lognormal distribution it might be useful to discuss the measures of the distribution (i.e. relationship of mean, median etc)

**Authors Response:**

The reported averages  $\mu$  are the arithmetic means and the corresponding standard deviation  $\sigma$ . The text has been updated to clarify that these are the values of the arithmetic means. We will also include a brief description of the lognormal distribution and the parameters that define it.

**21. Referee #2:**

p. 13, line 26: mm3
Authors Response:
The units of the volume are indeed mm3.

**22. Referee #2:**

p. 14, line 15 & p. 16, line 33: wow – fabulous. A porosity of 0.75 for 35 ppt agrees very well with estimates for frazil ice in layers under sea ice (called sub-ice platelet layer) (e.g. Langhorne et al, 2015).

**Authors Response:**

Thank you for highlighting this. We will include in the discussion section the agreement between our estimates of flocs porosities and the values reported by Langhorne et al (2015).

**23. Referee #2:**

p. 16, line 17-18: I'm not sure why the discrepancy between present measurements and those of Clark and Doering (2009) imply the latter are inadequate? Please explain. **Authors Response:**

Clark and Doering (2009) used a simplified criterion to identify individual flocs in the images, which was "a floc was considered to be any particle with an equivalent diameter greater than 17 mm" as quoted from their paper. Therefore, they disregarded any flocs smaller than this value and consequently they overestimated the mean sizes of flocs.

**24. Referee #2:**

Tables 2-5: Please reconsider rounding of all quantities. What is COV? Arithmetic mean or geometric mean (which I believe is equal to median)? For those not familiar with log normal distribution it might be useful to discuss the measures of the distribution (i.e. relationship of mean, median etc).

**Authors Response:**

Please see our response to comment number 3 and 20 above.

**25. Referee #2:**

Fig 1: clearly not a "Seabird". Would it be better labelled "temperature sensor"? Authors Response: Thank you for highlighting this, we updated the label to read "temperature sensor" as suggested.

**26. Referee #2:**

Fig 2 caption: "saline water in a confined vessel" to account for the decrease in freezing point.

**Authors Response:**

Good point. The text has been updated.

**27. Referee #2:**

Figs 13-15: What is the value of NT in each figure? Mark the means on the distributions by vertical lines.

**Authors Response:**

This is a good suggestion and we will add the values of NT and the arithmetic means to the captions for each figure.

The authors wish to thank Referee #3 for the constructive comments and corrections to the discussion paper. We have responded to each of the comments from the reviewer. The comments from the reviewer are in black font while our responses are in red font.

**1. Referee #3:**

**General Comments**

The authors have conducted a very thorough analysis of the effect of various levels of salinity on frazil ice formation and flocculation. The manuscript is very well organized and written, and fits nicely within the scope of the journal. It fills a gap in the literature and will be well received by the river and sea ice researchers. I recommend that the paper be accepted.

The introduction provides the reader with a good appreciation of the current state of knowledge and clearly outlines the contribution of this paper. Throughout the paper the authors do a good job of comparing with previous experiments and field measurements. and highlighting similarities and differences.

The experimental setup is described well, and the authors have conducted an impressive 9 runs of each test condition to evaluate the reproducibility of the experiments. The data is presented well and the analysis is thorough. The conclusions are supported by the data. The entire study is tied up quite nicely by the end. **Authors Response:**

Thank you for your positive comments and recommendation.

**2. Referee #3:**

**Specific Comments**

Page 2, line 29. Am I correct that the authors are implying that this polynya is always there? I understand that polynya's are persistent, but is this one permanent? Should also be 'produces' rather than 'produce'.

**Authors Response:**

Thank you for highlighting this. The sentence now is reworded to read "In the polynya that forms in the Chukchi Sea off Cape Lisburne and Point Hope, Alaska, rapid ice growth occurs and is accompanied by an ocean salt flux that produces a dense outflow... etc". This will clarify that this polynya does form intermittently in this region.

**3. Referee #3:**

Page 3, line 12 – might be nice to report the salinities of the Ushio and Wakatsuchi study rather than just saying low and high salinities. **Authors Response:**  We agree with this comment and the sentence has been reworded to read:" Ushio and Wakatsuchi (1993) performed laboratory experiments investigating the effect of wind conditions and salinity on the properties of frazil ice particles. In these experiments, the wind speed varied between 2 and 6 m/s, and water salinity between 0 and 35 ‰. They found that under strong wind and high salinity (above 25 ‰) conditions dendritic crystals having diameters of 2 to 3 mm were produced, and under calm air and lower salinity (below 25 ‰) conditions disc shaped crystals with diameters of approximately 5 mm were produced".

**4. Referee #3:**

Page 3, line 19 – Frazil 'deposits'. Line 22 – '44 ‰ water' should be reworded. Interchangeable use of disc and disk. As well as inconsistent hyphenation disc-shaped vs disc shaped, disk-like, etc.

**Authors Response:**

Page 3, line 19 now read:" Frazil deposits".

Line 22 now reads: "Hanley and Tsang (1984) performed laboratory experiments in a tank with water of 44 ‰ salinity and used a propeller to create turbulence". Thank you for highlighting the inconsistency in these terms. Page 4, line 1 reads:" disc-like". Also, the manuscript has been reviewed to avoid this inconsistency.

**5. Referee #3:**

If space is limited, Figure 2 could be removed. The experimentally-measured supercooling plot in Fig. 3 would be sufficient, and the 'theoretical' declining temperature of the residual supercooling level could just be described. COV is only defined in a figure caption.

**Authors Response:**

Thank you for your suggestion. We will consider removing Figure 2 if we are out of space. The COV is now defined in the text in Page 7, line 22.

**6. Referee #3:**

Page 9, line 31 – should be Eq. (2) rather than 1. Authors Response: Updated.

**7. Referee #3:**

The volume of particles being negligible compared to flocs is interesting, and a reader might question this based on the number of particles in Fig. 6 for example. Perhaps consider summing up the number and volume of particles and flocs in Fig. 6 and reporting them to help support your statement.

**Authors Response:**

We agree that the fact that "the volume of particles is negligible compared to flocs" was not accurately addressed in the text. We've now updated the text on Page 10, line 2 to read:" In all salinities, it was found that the total volume of frazil ice particles during peak concentrations was between 0.3 and 1% of the total volume of frazil ice flocs.

Therefore, the volume of frazil ice particles was neglected when computing ice volumes".

**8. Referee #3:**

Page 11, line 10 – the eccentricity statements could appear near Eq. (1). Authors Response:

Updated, the sentence:" Ellipses have eccentricity between zero and one, where an eccentricity of zero corresponds to a circle and an eccentricity of one corresponds to a straight line." has been moved to Page 9 line 6.

**9. Referee #3:**

At a glance, Figure 9 and 10 show numbers and sizes of frazil particles that don't appear to be much different than those of the frazil flocs. Have the size distributions plotted separately for Figs. 13 and 14 also make it difficult to compare. Consider combining these two figures.

**Authors Response:**

A careful examination at Figure 9a shows an average peak concentration of about 2 particle/cm3, as opposed to ~ 1 floc/cm3 from Figure 10a. We agree that Figures 13 and 14 might be difficult to compare especially since the x-axis is a log scale and has different limits in the two figures. We adjusted Figure 13 to have x-axis limits of 0.01 and 100 (mm) to match the limits in Figure 14 and this should facilitate comparison. We think this is a better solution than superimposing these two figures together since this would obscure the individual bars in each distribution and would make it difficult for the reader to distinguish between them.

**10. Referee #3:**

Page 14, Line 9 – Latent heat of fusion of ice seems to be in J/g, rather than J/kg. The discussion sometimes switches from past to present tense in a way that doesn't seem to always be correct.

**Authors Response:**

Thank you for highlighting this. The value of the latent heat of fusion of ice is now updated to  $3.33 \times 10^5$  J/kg. Also, the past tense has been used throughout this discussion for consistency.

[revised manuscript text omitted]
 and flocs mean sizes and concentrations along with simultaneously measured supercooling temperatures observed at different salinities are being-presented for the first time. The effects of different salinities on the characteristics of particles and flocs is highlighted, and estimates of flocs porosities are presented. Measurements of the size and shape of individual frazil ice particles

and frazil ice flocs in saline and freshwater can be applied-used to improve river ice models (e.g. Shen, 2010) and sea ice

**25 2 Experimental Set-up and Methods**

Experiments were performed in the frazil ice production tank in the University of Alberta's Cold Room Facility as described in Ghobrial et al. (2012). An image of the tank and the experimental set-up is presented in Fig. 1. The tank has a base dimension of 0.8 m by 1.2 m and was filled to a depth of 1.2 m. The four bottom-mounted propellers used to generate turbulence in the tank are driven by four NEMA 34 DC variable speed electric motors (278 W, 1.514 N-m of torque, max speed of 1750 rpm).

30 The turbulence intensity was held constant by keeping the propeller speed constant at 325 rpm for all experiments. The speed of each motor was verified using a laser tachometer. In a similar series of experiments in the same tank, McFarlane et al. (2015) found that the tank-averaged turbulent kinetic energy dissipation rate was  $\sim 3.436 \times 10^{12}$  -em2/s3 at a propeller speed of 325

rpm, and this fell within the range of dissipation rates estimated for rivers in Alberta. In general, the dDissipation rates in the oceans range from ~  $10^{,2}$  m2/s3 to  $10^{,9}$  m2/s3 (Banner and Morrison, 2018; Wang and Liao, 2016) with a reported lower range in the polar regions ranging from ~  $10^{,3}$  m2/s3 to  $10^{,10}$  m2/s3 (Chanona et al., 2018; Scheifele et al., 2018). Therefore, the dissipation rate generated in these experiments falls within the upper limits of values observed in the oceans. Future

experiments would be valuable forneeded to investigateing the behaviour of frazil formation at lower dissipation rates.

5

Two types of light-emitting diode (LED) lights were used in the experiments to illuminate the frazil ice particles: a Genaray SpectroLED Essential 360 Daylight LED Light (3,200 lux at 1.0 m, 360 LED bulbs, 29.8 cm by 29.8 cm) and an Andoer FalconEyes RX-18TD 504 pcs LED Light (3660 lux at 1 m, 504 LED bulbs, 70.0 cm by 46.0 cm). The light source was placed on the far side of the tank (Fig. 1). On the opposite side of the tank, two Cavision glass polarizers were mounted in the tank flush to the front glass wall. Two different polarizer sizes were used in the experiments, either a 10 by 10 cm and-or a 16 by 16 cm polarizers. In both cases, the polarizers were installed at 90° to one another in order to cross polarize the light. This setup produced a black background in the captured images where only the ice particles / flocs passing between the two polarizers, that had refracted the incident light, were visible. The polarizers were mounted as close as possible to the front glass sidewall to prevent any distortion of the images caused by suspended frazil ice getting between the sidewall and the polarizers. Digital images of frazil ice and flocs were captured using a Nikon D800 camera (36-megapixel resolution) equipped with an AF Micro-Nikkor 60 mm f/2.8D lens. The camera was positioned outside of the tank and focused on the polarizers

(see Fig. 1). A space heater was used to blow hot air against the outside of the glass sidewall to prevent frost formation.

- 20 Based on some preliminary experiments, three different camera and polarizer set-ups that provided the best quality images, with regards to brightness and clarity were determined and these are summarized in Table 1. For Set-up 1, it was determined found that in freshwater the 2.2 cm spacing between the polarizers was appropriate to capture individual frazil particles but prevented many flocs from advecting between the polarizers and also the flocs were sometimes too large for the 47.5 by 31.7 mm field of view. For Set-up 2, the flocs in saline water were observed to be small enough that a 2.2 cm spacing between the
- 25 polarizers did not restrict their movement and the flocs were small enough to fit in the slightly larger field of view of 61.3 by 40.9 mm. In the case of Set-up 3, a 3.5 cm spacing between the polarizers and a larger field of view of 162.9 by 141.3 mm were needed to accommodate the larger freshwater flocs. The three different set-ups were used during the five series of experiments listed in Table 2.
- 30 The temperature of the water in the tank during the experiments was recorded at a rate of 0.62 Hz using a Sea-Bird SBE 39 temperature recorder (accuracy of ± 0.002 °C). The temperature sensor was mounted in the tank and placed at the approximate centre of the tank. Temporal variations of air temperature in the cold room were measured using RBR Solo Temperature Loggers (accuracy of ± 0.002 °C) at a frequency of 1 Hz. A series of experiments were performed in freshwater to determine if the water in the tank was well mixed and if the water temperature was uniform in the tank at a propeller speed of 325 rpm.

[revised manuscript text omitted]

| Salinity   | Mean sizes ± standard deviations (mm)
Overall |                 |                 |                 |
|------------|--------------------------------------------------|-----------------|-----------------|-----------------|
|            | Phase 1                                          | Phase 2         | Phase 3         | Overall  |
| Freshwater | $0.54 \pm 0.58$                                  | $0.58 \pm 0.46$ | $0.48 \pm 0.33$ | $0.52 \pm 0.41$ |
| 15 ‰       | $0.54 \pm 0.43$                                  | $0.50 \pm 0.36$ | $0.42 \pm 0.30$ | $0.46 \pm 0.35$ |
| 25 ‰       | $0.53 \pm 0.35$                                  | $0.49 \pm 0.34$ | $0.44 \pm 0.31$ | $0.48 \pm 0.33$ |
| 35 ‰       | $0.50 \pm 0.32$                                  | $0.47 \pm 0.32$ | $0.42 \pm 0.29$ | $0.45 \pm 0.31$ |

Table 3: Mean Arithmetic mean sizes and corresponding standard deviations of frazil ice particles during different phases and at all four salinities.

Table 4: Mean Arithmetic mean sizes and corresponding standard deviations of frazil ice flocs during different phases and at all four salinities.

| Salinity   | Mean sizes ± standard deviations (mm)
Overall |                 |                 |                 |
|------------|--------------------------------------------------|-----------------|-----------------|-----------------|
|            | Phase 1                                          | Phase 2         | Phase 3         | Overall  |
| Freshwater | $1.68 \pm 1.19$                                  | $2.28 \pm 2.06$ | $2.65 \pm 3.09$ | $2.57 \pm 2.88$ |
| 15 ‰       | $0.93 \pm 0.96$                                  | $1.45 \pm 1.30$ | $1.83 \pm 1.81$ | $1.64 \pm 1.63$ |
| 25 ‰       | $1.02 \pm 0.81$                                  | $1.39 \pm 1.09$ | $1.77 \pm 1.57$ | $1.61 \pm 1.43$ |
| 35 ‰       | $0.96 \pm 0.82$                                  | $1.30 \pm 1.01$ | $1.60 \pm 1.40$ | $1.47 \pm 1.28$ |

Table 5: Sizes of the largest frazil ice flocs at all four salinities.

| Salinity 95 th Percentile (mm) |      | Mean Size of Flocs Larger than
95 th Percentile (mm) | Maximum Floc Size
(mm) |
|-------------------------------------------|------|--------------------------------------------------------------------|---------------------------|
| Freshwater                                | 6.91 | 11.89                                                              | 95.10                     |
| 15 ‰                                      | 4.82 | 6.77                                                               | 36.22                     |
| 25 ‰                                      | 4.38 | 5.98                                                               | 23.18                     |
| 35 ‰                                      | 3.96 | 5.38                                                               | 25.19                     |

| Salinity   | Mean volumes ± standard deviations (mm 3 )
Overall |                  |                    |                 |
|------------|------------------------------------------------------------------|------------------|--------------------|-----------------|
|            | Phase 1                                                          | Phase 2          | Phase 3            | Overall  |
| Freshwater | $0.40 \pm 1.07$                                                  | $3.01 \pm 37.68$ | $10.67 \pm 141.45$ | 8.79 ± 117.98   |
| 15 ‰       | $0.21 \pm 4.70$                                                  | $0.60 \pm 4.36$  | $1.54 \pm 7.78$    | $1.14 \pm 6.68$ |
| 25 ‰       | $0.15 \pm 0.71$                                                  | $0.39 \pm 1.49$  | $1.08 \pm 4.55$    | $0.82 \pm 3.78$ |
| 35 ‰       | $0.16 \pm 2.62$                                                  | $0.31 \pm 1.05$  | $0.78 \pm 3.22$    | $0.60 \pm 2.72$ |

 Table 6: Estimated arithmetic mean volumes and corresponding standard deviations of frazil ice flocs during different phases and at all four salinities.

**Figures**